# Pump-probe X-ray holographic imaging of laser-induced cavitation bubbles with femtosecond FEL pulses

M. Vassholz [1], H. P. Hoeppe [1], J. Hagemann [2], J. M. Rosselló [3], M. Osterhoff [1], R. Mettin[3], T. Kurz[3], A. Schropp[2], F. Seiboth [2], C. G. Schroer [2,4], M. Scholz[5], J. Möller[5], J. Hallmann[5], U. Boesenberg[5], C. Kim[5], A. Zozulya [5], W. Lu[5], R. Shayduk[5], R. Schaffer[5], A. Madsen [5] & T. Salditt [1✉]

Cavitation bubbles can be seeded from a plasma following optical breakdown, by focusing an intense laser in water. The fast dynamics are associated with extreme states of gas and liquid, especially in the nascent state. This offers a unique setting to probe water and water vapor far-from equilibrium. However, current optical techniques cannot quantify these early states due to contrast and resolution limitations. X-ray holography with single X-ray free-electron laser pulses has now enabled a quasi-instantaneous high resolution structural probe with contrast proportional to the electron density of the object. In this work, we demonstrate cone-beam holographic flash imaging of laser-induced cavitation bubbles in water with nanofocused X-ray free-electron laser pulses. We quantify the spatial and temporal pressure distribution of the shockwave surrounding the expanding cavitation bubble at time delays shortly after seeding and compare the results to numerical simulations.

[1] Institut für Röntgenphysik, Georg-August-Universität Göttingen, Göttingen, Germany. [2] CXNS - Center for X-ray and Nano Science, Deutsches Elektronen-Synchrotron DESY, Hamburg, Germany. [3] Drittes Physikalisches Institut, Georg-August-Universität Göttingen, Göttingen, Germany. [4] Department Physik, Universität Hamburg, Hamburg, Germany. [5] European X-Ray Free-Electron Laser Facility, Schenefeld, Germany. ✉email: tsaldit@gwdg.de

S mall transient or strongly driven cavitation bubbles in liquids exhibit a wide range of interesting nonlinear effects. They can experience violent collapse[1,2], which is associated with shockwave emission into the liquid, high compression, heating of the bubble medium, light emission (sonoluminescence) or chemical reactions. In the vicinity of a solid surface or interface they can form liquid jets, resulting in erosion of the material. In ultrasonically driven multi-bubble systems (acoustic cavitation)[3], the mutual interaction of bubbles and their interaction with the sound field can lead to structure formation and collective behavior. Apart from fundamental aspects of non-equilibrium physics, these processes are relevant for a range of medical procedures, for example to emulsify tissue in cataract surgery[4] or bubble-mediated drug delivery[5]. The understanding of cavitation bubbles and dynamics is important as well for sonochemistry, ultrasonic cleaning and corrosion prevention. For well-controlled experiments on cavitation bubbles, short laser pulses are commonly used, which seed cavitation bubbles by the transition from a laser-generated plasma to a hot, compressed bubble nucleus, and finally to an expanding gas and vapor bubble in the liquid environment. This transition from the plasma to a bubble, the plasma growth, subsequent cooling of the plasma and generation of shockwaves in the medium, as well as the precise states of matter in the bubble remain elusive. For several decades, the main tools to study cavitation dynamics have been acoustic methods, optical pump-probe spectroscopy[6] and optical imaging[7], with up to 100 million frames per second by high-speed ICCD cameras[8]. Increasing sensitivity of optical sensors has more recently allowed for direct imaging of bubble oscillations and sonoluminescence light emission in multi-bubble fields[3]. Likewise, the initial bubble formation and shockwave emission after dielectric breakdown was measured with acoustical methods and optical methods, such as bright and dark field imaging, optical interferometry, Schlieren photography, and streak imaging[9–16]. However, due to the small scales and the fast dynamics, imaging of the bubble interior and its close environment during dielectric breakdown and collapse still poses unmet challenges. Optical methods are limited by the numerical aperture of long-distance objectives, required to image cavitation bubbles sufficiently far from interfaces. Sub-nanosecond time resolution and sub-micrometer spatial resolution are required to follow the motion of the phase boundary and the dynamics of the bubble interior. In the absence of direct imaging methods, knowledge of the collapsed bubble state has been inferred from spectroscopic measurements of the emitted light[17], and has been based on model calculations[18–21]. Several models have been developed to describe the nonlinear phenomenon of dielectric breakdown in liquids and the following cavitation dynamics[22–27]. However, many aspects of the dynamical evolution of the bubble and the structure of the phase boundary remain unclear. Open questions relate to, e.g., the presence of inhomogeneities, the existence of converging shocks, and even more fundamentally to the exact spatial density and pressure profile of the bubble and the surrounding shockwave in different states.

In this work, we demonstrate near-field holographic imaging of cavitation bubbles with single X-ray free-electron laser (XFEL) pulses. This experimental approach offers a quasi-instantaneous high resolution structural probe at different stages after seeding, particularly useful to investigate extreme states of bubble generation and collapse. The method offers higher resolution and penetration depth than ultra-fast optical microscopy, and importantly a unique direct sensitivity to the electron density profile, which is not accessible by the aforementioned optical methods. Such experimental data are required to assess the validity and limits of current numerical models and theoretical hypotheses and improve our basic physical understanding of these processes. More generally, near-field X-ray holography with nano-focused single FEL pulses is a promising tool to study driven condensed matter and warm dense matter. Cone beam holography with XFEL pulses was previously used to image shockwave propagation in diamond[28]. In contrast to the shockwave propagation in solids, we image the dynamics of complex phase transitions in liquid water after dielectric breakdown, with higher geometrical complexity. Compared to the recently demonstrated X-ray microscopy of laser-induced dynamic processes with parallel beam optics[29,30] or an incoherent plasma X-ray source[31], the present method offers higher spatial resolution and sensitivity, not limited by the detector pixel size. We have measured micrometer-sized cavitation bubbles in a pump-probe imaging scheme with single XFEL pulses. For a quantitative analysis, we have developed a high-throughput workflow of the geometrically magnified near-field holograms. To this end, we introduce a phase retrieval approach, which makes use of the radial symmetry of the cavitation bubbles. With this analysis, the three dimensional (3d) mass-density distribution of the bubble's interior, of the interface between bubble and shockwave, as well as of the shockwave surrounding the cavitation bubble is obtained at a spatial sampling of about 100 nm pixel size and a temporal resolution of a few nanoseconds, only limited by the pulse duration of the pump laser. The density profiles allow to extract the 3d-pressure distribution of individual shockwaves in space and time in close proximity to the cavitation center. This pressure distribution is not accessible with other methods. Optical methods only measure a single pressure value directly at the shockfront[9], leaving the pressure distribution in between bubble and shockfront unknown. Hydrophones for acoustic methods cannot be placed in close proximity to the cavitation center. We compare the measured pressure distribution with simulations based on the commonly used Gilmore-Akulichev model for cavitation[32]. In total, density and pressure distributions are evaluated for more than 3000 individual cavitation events, which can then be used to compute histograms of physical properties beyond simple ensemble averages.

## Results

**Instrumentation and implementation**. To observe cavitation dynamics with X-ray near-field holography (NFH), an infrared (IR) laser-pump and X-ray-probe scheme is employed. The main components of the experimental setup (Fig. 1a) are the focusing optics of the X-ray beam, a pulsed IR laser generating cavitation inside a water-filled cuvette and an X-ray camera recording the X-ray holograms. The experiment is performed at the MID (Materials Imaging and Dynamics) instrument[33,34] of the European XFEL[35]. The XFEL provides ultra-short X-ray pulses on the order of 100 fs, or less, with a photon energy of 14 keV at a repetition rate of 10 Hz and $3 \times 10^{11}$ photons in average per pulse. X-rays are focused with a set of Beryllium compound refractive lenses to a focal spot size of ~78 nm (calculated full width at half maximum, FWHM)[36]. A focused IR laser with wavelength 1064 nm, numerical aperture 0.2 , 6 ns pulse duration and 24 mJ pulse energy, excites cavitation events inside a water-filled cuvette. The cuvette is placed in a distance of $z_{01} = 144$ mm behind the X-ray focus. The holographic contrast is formed by free-space propagation towards the scintillator-based (LuAg:Ce, thickness 20 μm) X-ray camera positioned at a distance of $z_{02} = 9578$ mm behind the X-ray focus. The geometric magnification of $M \approx 66.5$ yields an effective pixel size in the sample plane of $d_{eff} = 98$ nm and a Fresnel number of $F = 7.6 \times 10^{-4}$. The setup is operated in air, but an 8 m long evacuated flight tube between the setup and X-ray camera reduces absorption losses. The X-ray data is complemented with additional measurements. A high-speed (HS)

optical camera observes the cavitation process simultaneously to the X-rays (Fig. 1a, c). The acoustic signal of the cavitation events is recorded by a piezo-ceramic microphone glued to a wall of the cuvette. The following measurement scheme is operated with 10 Hz repetition rate (Fig. 1b): (i) The IR pump laser shoots into the water-filled cuvette inducing a cavitation bubble with probability $\eta$. (ii) After a time delay $\Delta t$ the FEL X-ray pulse probes the excited bubble and the X-ray camera records the hologram. (iii) The HS optical camera records multiple frames, where the first frame is synchronized to the IR laser pulse to detect the plasma spark. (iv) A digital oscilloscope records the signal of the microphone. The cavitation dynamics are recorded by measurements for different time delays $\Delta t$ between IR laser pump and X-ray probe (Fig. 1d). Details on the experimental setup and the timing scheme are given in the Methods section and in[37].

With this measurement scheme, we acquired X-ray holograms for more than 20,000 individual cavitation events. To extract the quantitative phase of the cavitation bubbles from the holograms, we present a tailored phase retrieval approach for objects with radial symmetry. The phase retrieval gives access to physical quantities of the cavitation bubbles and enables to resolve the density and pressure in space and time. In the following we analyze single individual cavitation events, followed by an automated procedure to extract phase, density and pressure individually for an ensemble of over 3000 cavitation events. The automated selection was carried out based on criteria to ensure that the hologram contained a single cavitation bubble only, which did not exceed the field of view. Based on the spatial density and pressure distributions, we show how key properties of the cavitation dynamics change with the deposited laser energy. If not stated otherwise, we always refer to the shockwave generated by the dielectric breakdown rather than the shockwave emitted by the bubble collapse.

**Phase retrieval reveals the bubble density profile.** Near-field holographic X-ray imaging encodes the object's phase shift and absorption properties in intensity modulations based on self-interference of the undisturbed primary beam and its modulations by the sample. Phase retrieval denotes the process of decoding the sample's properties from the intensity measurements, i.e., the hologram. In a first pre-processing step, contributions of an imperfect illuminating wavefront have to be identified and removed. In synchrotron experiments this is typically done by a simple empty-beam division, i.e., dividing the measured intensity with sample by the intensity of the empty beam. This approach requires stable beam properties. However, the spontaneous nature of the SASE process of FEL radiation leads to strong pulse to pulse fluctuations, including strong variations in the total intensity and pointing of the X-ray beam, impeding empty-beam correction. To overcome these challenges, we acquire a set of single-pulse empty beams and decompose this set into its statistical contributions by a principal component analysis (PCA). The best suited linear combination of components is determined for each single-pulse hologram individually and used for empty-beam correction. This approach was initially proposed for synchrotron data[38] and is described in more detail for FEL radiation in[39].

A variety of phase retrieval algorithms are available, including single step[40] and iterative approaches[39,41,42]. Here, we use a phase retrieval approach, which exploits the radial symmetry of the cavitation bubbles to reduce complexity and requirements on the signal-to-noise ratio of the measured holograms. We denote this approach the Radially Fitted Phase (RFP). RFP is a forward-model approach, minimizing the difference between the measured intensity and the numerically propagated intensity of the sample's phase shift $\overline{\phi}$, as illustrated in Fig. 2a–c. The radial intensity $I_{\mathrm{meas}}(R)$ is calculated by averaging over the polar angle of an empty-beam corrected and center-shifted hologram (Fig. 2a, b). The phase retrieval approach is formulated as an optimization problem, searching for the projected radial phase $\overline{\phi}(R)$ (Fig. 2c) minimizing the $\ell^2$-norm between the numerically forward propagated intensity $I(\overline{\phi})$ and the measured radial intensity $I_{\mathrm{meas}}$. A fast and efficient Hankel-transform based Fresnel-type propagator is used for the propagation in radial coordinates. Furthermore, we exploited the fact that the stoichiometry of water in the cuvette is constant, albeit at different density, i.e., our sample consists of a single material with non constant complex-valued index of refraction $n(R) = 1 - \delta(R) - i\beta(R)$, but with constant ratio $\beta/\delta$. Details on the propagator, the optimizer, and the calculation of the center coordinates of the cavitation bubble are given in the "Methods" section. For comparison, Fig. 2d shows the two-dimensional projected phase, retrieved by the iterative Alternating Projections (AP) scheme[43]. The polar angle average of the AP reconstruction is compared to the RFP reconstruction in Fig. 2e.

The phase retrieval gives access to the projected phase $\overline{\phi}$. However, to obtain information on the 3d-density distribution of the cavitation bubble, a projection inversion is needed. Assuming

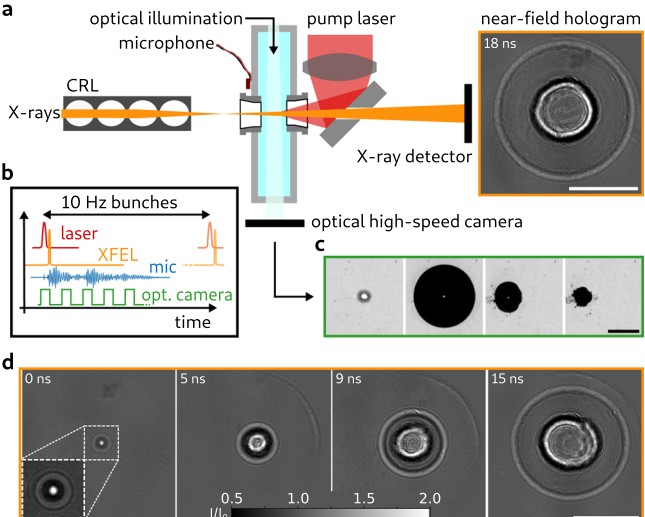

**Fig. 1 Holographic imaging of cavitation at the MID instrument. a** The FEL X-ray pulses are focused to nanometer spot size by the beryllium CRLs. A cuvette with water is placed behind the X-ray focus. The pump laser is focused by a lens and reflected by a subsequent plane mirror into the water to seed the bubble. The X-ray and the laser beam are antiparallel. The X-ray beam passes through a small hole in the laser mirror to the X-ray detector. The distance between X-ray focus and laser focus, i.e., the seeding point of cavitation, is $z_{01} = 144$ mm and between X-ray focus and detector $z_{02} = 9578$ mm. A high-speed optical camera observes the bubble formation perpendicular to the X-ray beam. A microphone at the cuvette's wall registers the acoustic signal of cavitation events. **b** Timing scheme of the experiment. The pump laser excites a cavitation bubble at a time $\Delta t$ prior to the FEL pulse. The optical high-speed camera acquires a series of images with the first frame synchronized to the pump laser pulse. The microphone signal of the acoustics is recorded (mic). **c** Image sequence of the optical high-speed camera. The first frame (left) shows the plasma spark. The following frames have time delays of 40 μs, 140 μs, and 160 μs (left to right) with respect to the first frame. **d** Empty-beam corrected X-ray holograms of cavitation events at different times $\Delta t$, indicated in the top left corner. The holograms show strong contrast at the inner interface (gas/shockwave) and at the outer interface (shockwave/equilibrium water). Scale bars: 50 μm (**a**, **d**), 500 μm (**c**).

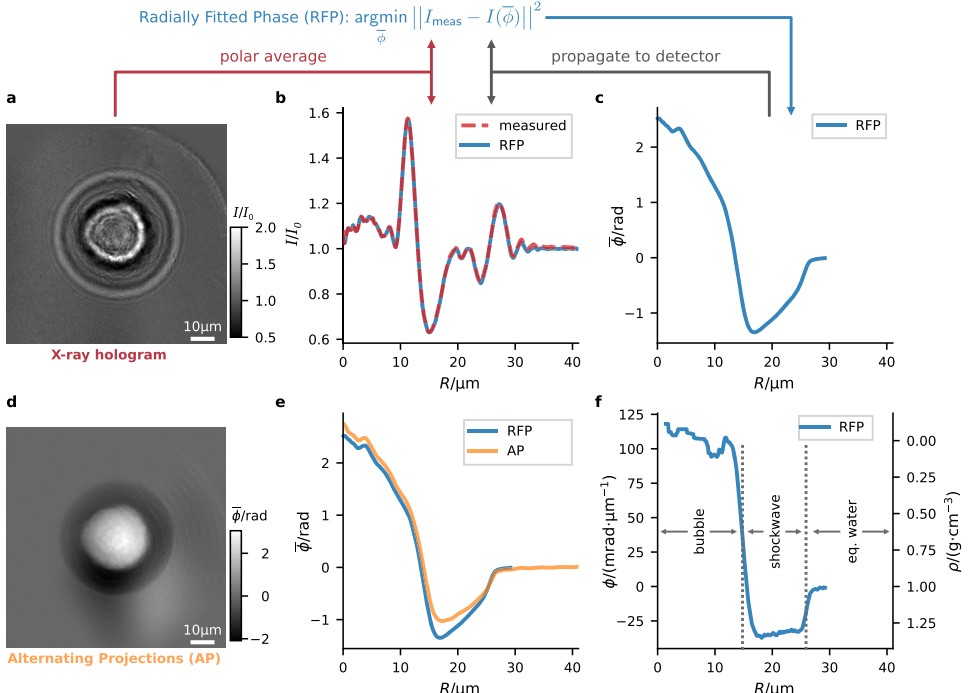

**Fig. 2 Holographic phase retrieval and cavitation bubble density. a** X-ray hologram (normalized intensity $I/I_0$) of a cavitation bubble at $\Delta t = 10$ ns, exhibiting strong contrast at the inner interface (gas/shockwave) and outer interface (shockwave/equilibrium water). For phase retrieval, the hologram is averaged along the polar angle to obtain the radial intensity distribution. **b** Radial intensity distribution of (**a**) and intensity obtained from numerical propagation of the RFP retrieved phase (see (**c**)). **c** In a forward model approach the projected phase $\overline{\phi}$ of the bubble is retrieved by minimizing the difference to the radial intensity distribution (Radially Fitted Phase, RFP). **d** Retrieved phase of (**a**) using the AP algorithm, for comparison. The phase distribution reflects the deficit density in the core and excess density in the shockwave. **e** The average along the polar angle of the AP reconstruction is compared to the result obtained from RFP (**c**). **f** Radial three dimensional phase $\phi$ reconstructed from the RFP projected phase (**c**). The right ordinate shows the calculated density distribution of the cavitation bubble for an ellipticity factor $\epsilon \approx 0.8$. Scale bars: 10 μm (**a**, **d**).

sphericity of the bubble, the projection inversion is given by the inverse Abel transformation. We use a regularized version of the inverse Abel transform, which stabilizes the inner voxels with low volumetric weight against noise (see "Methods"), to obtain the 3d phase shift $\phi(R)$ of the cavitation bubbles (Fig. 2f). The measured phase describes the difference of the sample to the surrounding medium, which is in this case water at equilibrium. Thus, a positive/negative phase shift corresponds to an electron density lower/higher than uncompressed water, respectively. $\phi(R)$ describes the phase shift induced per voxel as a function of distance $R$ to the center of the bubble and is proportional to the mass density $\rho(R)$ at a given distance $R$ as $\rho(R) = \rho_0(1 - \epsilon\,\phi(R)/(k\,\delta))$, with $k$ being the wavenumber of the X-rays and $\rho_0 \simeq 1$ g cm$^{-3}$ the equilibrium density of water. We determine the radius of the bubble boundary $R_B$ and shock front $R_{SW}$ at the FWHM of the respective slope of the density profile. These key values are indicated by the vertical dotted lines in Fig. 2f. To compensate for an initial ellipticity of the cavitation bubble, originating from a plasma elongation in the direction of the laser during dielectric breakdown, we introduce an ellipticity factor $\epsilon$ to relax the constraint on sphericity to axisymmetric ellipsoidal bubbles. We define the ellipticity factor to be the ratio of the two principal axes of the ellipsoid $\epsilon = a_\perp/a_z$, where $a_z$ is the principal axis along the direction of the X-ray beam and $a_\perp$ the principial axis perpendicular to the beam. $\epsilon$ is chosen such that the density of the vapor inside the bubble cavity corresponds to the density of water vapor $\rho \approx 0$. Figure 2f shows the phase profile (left axis) and density profile (right axis) of an exemplary cavitation bubble, consisting of gas phase core (phase maximum/density minimum) and shockwave shell (phase minimum/density maximum). For this bubble, the ellipticity factor evaluates to $\epsilon \approx 0.8$. The

shockwave exhibits a density excess of ~0.3 g/cm$^3$. The ellipticity of the bubble changes quickly with the time delay $\Delta t$ (cf. Supplementary Fig. 1c). The median of the ellipticity decreases to the minimum value of $\epsilon \approx 0.7$ within the first ~6 ns and relaxes to 0.9–1 at ~18 ns.

**Pressure distribution.** Based on the mass density $\rho(R)$ we calculate the spatial pressure distribution $p(R)$ of the shockwave using the empirical Tait equation of state[44]

$$\frac{p(R) + B}{p_0 + B} = \left(\frac{\rho(R)}{\rho_0}\right)^n, \qquad (1)$$

with the hydrostatic pressure $p_0 = 0.1$ MPa and the constants $B = 314$ MPa and $n = 7$ for water[45]. Figure 3 shows the 3d radial phase distribution $\phi(R)$ and the pressure distribution of the shockwave $p(R)$ for three different time delays, without ellipticity correction. For each $\Delta t$ two different bubble energies $E_B$ are shown. The energy of the cavitation event was estimated from the bubble lifetime $\tau$, i.e., the time between dielectric breakdown and collapse, measured by the signal of the microphone at the cuvette's wall. The energy driving the bubble $E_B$ scales approximately linearly with the third power of the lifetime $\tau$[46] (see "Methods" for details). Figure 3 demonstrates that with X-ray holography the pressure distribution of the shockwave $p(R)$ can be obtained in close proximity to the center of the cavitation event. The cavitation events with high bubble energy $E_B$ show an initial peak pressure of more than 20 GPa, the low energy events have peak pressures ~10 -times lower. Note that we have some uncertainty in the pressures due to the exact shape of the bubble along the projection direction (X-ray beam axis). It is certainly reasonable to assume axial symmetry, and we can also correct for an

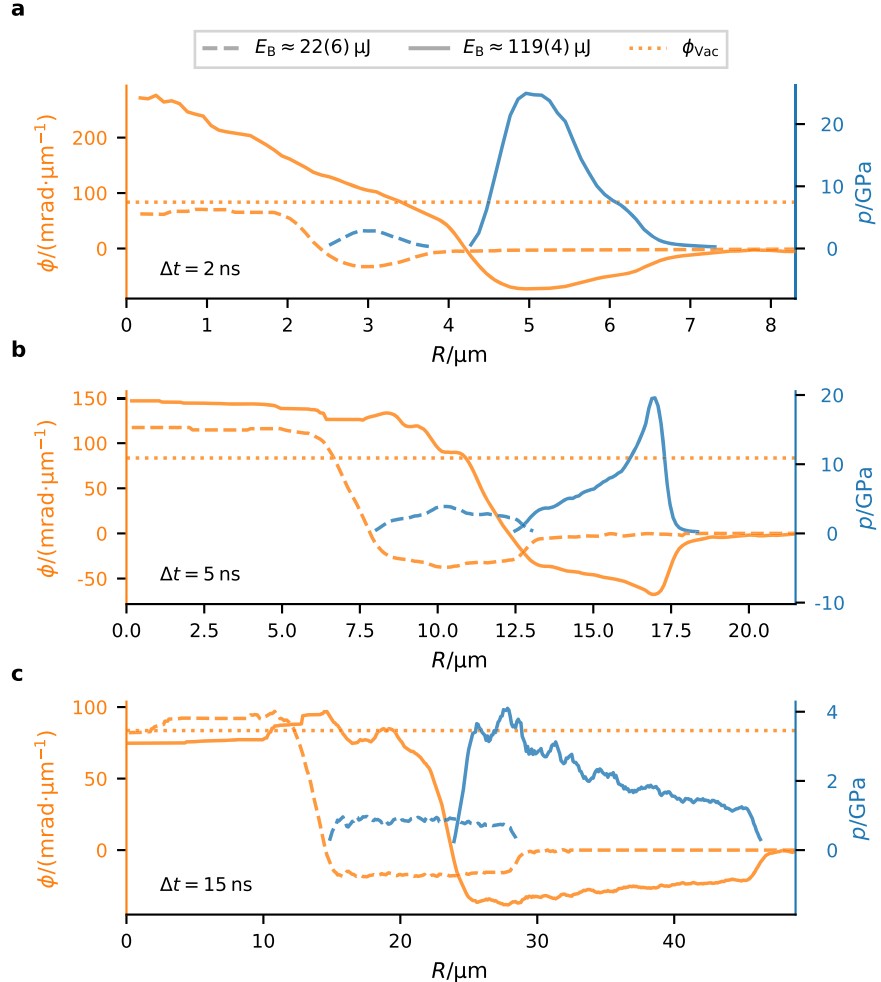

**Fig. 3 Phase and pressure distributions of individual bubbles. a–c** Radial phase $\phi(R)$ and spatial shockwave pressure $p(R)$ for $\Delta t = 2$ ns, 5 ns and 15 ns, respectively. For each delay two exemplary cavitation events with energy of $E_B \approx 22(6)$ μJ (dashed) and $119(4)$ μJ (solid) are compared. The 3d-phase distribution $\phi(R)$ is shown on the left ordinate (orange), the pressure distribution of the shockwave $p(R)$ on the right ordinate (blue). The phase shift of vacuum to water $\phi_{vac}$ (dotted) is shown for comparison. A phase profile exceeding this line (as is typically the case for small $\Delta\tau$ and high $E_B$) indicates a non-spherical bubble, and hence the necessity to introduce the ellipticity factor $\epsilon$ (see text). The pressure distribution of the shockwave was calculated using the Tait equation.

ellipsoidal shape, as discussed above. However, higher order contributions (in particular cone- or pear-like distortions) may also be present. This would, however, not affect the overall features of the extracted distribution such as the sign of the pressure slope or the width of the pressure distribution. Before we compare the obtained pressure distribution with simulated data, we will have a closer look at the dynamics of cavitation bubbles and the shockwave pressure in the next part.

**Density and pressure dynamics.** Out of 20,000 holograms of individual cavitation events, we processed an automatically selected subset of over 3000 events. For each event the 3d-spatial phase distribution was retrieved. A summary of the results is shown in Fig. 4. The evolution of the bubble boundary radius $R_B$ (radius of the interface bubble to shockwave) and the shockwave radius $R_{SW}$ (outer boundary of shockwave to equilibrium water) shows a faster decrease of bubble wall velocity for lower energetic cavitation events (Fig. 4a). Each of the scatter dots shown in Fig. 4a represents one cavitation event with an individually retrieved phase distribution $\phi(R)$. In the following, we narrow the data down to describe the density and pressure dynamics for different energy ranges of the ensemble. To this end, we process the median of the 3d phase shift $\phi_{med}(R)$ of all events of the

ensemble for which the bubble boundary radius $R_B$ and the energy values $E_B$ are within a specified range. Figure 4b shows the median of the phase shifts for cavitation events with $R_B$ between 2 and 3 μm. This step is repeated for different ranges of $R_B$ (Fig. 4c), color-coded with the median time delay $\Delta t$. Here, only cavitation events with energy $E_B$ between 66 and 130 μJ were used. From the envelope of the shockwave's phase shift (median profiles), we calculate the peak-pressure distribution of the shockwave $p_{peak}(R)$ as a function of the distance $R$ to the center of the cavitation event. This value describes the average peak pressure that an observer measures in a distance $R$ when the shockwave travels by. Figure 4d shows $p_{peak}(R)$ calculated from the median 3d phase profiles for three different energy ranges. Note, that here we did not compensate for ellipticity in the pressure calculation. Supplementary Figure 7 shows the same data with ellipticity correction. However, in this case the evolution of the peak pressure $p_{peak}(R)$ does not monotonically decrease after reaching its maximum. This hints at the fact that cone- or pear-like shape distortions are more important at these time scales [9]. In this case, the shockwave of the bubble is better modeled by a sphere than an ellipsoid, even if the cavity is not. We will see in the next section that the overall average pressure of the shockwave without ellipticity correction indeed fits reasonably well with the simulations.

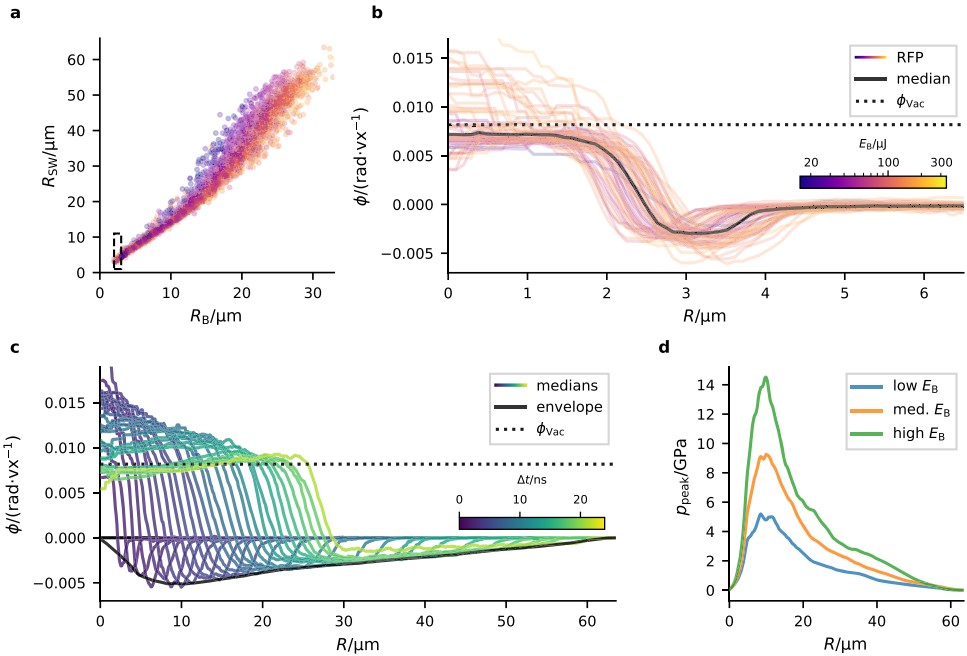

**Fig. 4 Cavitation dynamics. a** Radius of bubble and shockwave boundary $R_B$ and $R_{SW}$. Each scatter dot represents one processed cavitation event. The color scales with the bubble's energy (shared colorbar with (**b**), logarithmic scale). **b** Radial 3d phase profiles $\phi(R)$ of cavitation events with 2–3 μm bubble boundary radius (dashed box in (**a**)). The radial phase was reconstructed from the RFP phases $\overline{\phi}$. The color represents $E_B$. The median of all phase distributions is shown in black. The phase shift of vacuum to water $\phi_{vac}$ is shown for comparison. **c** Median of phase profiles for different ranges of $R_B$, showing how the median phase evolves with time. Here, only cavitation events with $E_B$ between 66 and 130 μJ were used. The color represents the median of the time delay $\Delta t$. The (smoothed) envelope of the shockwave's phase shift (black) is used to calculate the shockwave's peak pressure $p_{peak}$ as a function of the distance to the bubble center $R$. **d** $p_{peak}(R)$ obtained from the envelope of the shockwave's phase shift for energy ranges $E_B$ between 7 and 66 μJ (low $E_B$), 66 and 130 μJ (med. $E_B$) and 130–250 μJ (high $E_B$).

**Comparison to numerical simulations**. We will now compare our data to results obtained from numerical simulations using the Gilmore-Akulichev model[32] (in the following referred to as Gilmore model). The Gilmore model describes the dynamics of the bubble wall accounting for compressibility of the liquid and sound radiation. It allows the calculation of the shockwave, that is emitted during the rapid bubble expansion, via the Kirkwood-Bethe hypothesis[47]. Both steps use the modified Tait equation of state (1) for water (see "Methods" for further details).

For two exemplary energy ranges of the bubble energy $E_B$, we optimized the starting conditions of the simulations (similar as in[9]) to fit the trajectory of the bubble wall radius $R_B(\Delta t)$. The low $E_B$ simulation was optimized for data in the energy range $E_B$ between 20 and 33 μJ and the high $E_B$ simulation for 111–130 μJ. Figure 5a shows the trajectories $R_B(\Delta t)$ and $R_{SW}(\Delta t)$ for the high $E_B$ simulation together with the experimental values in the corresponding energy range (cf. Supplementary Fig. 9 for the low $E_B$ trajectories). The Tait equation overestimates the shockwave speed for shock pressures exceeding 2.5 GPa[23]. To compensate this overestimation in our simulations, we treat the value $B$ as an effective parameter of the Tait equation. With an adjustment (see Supplementary Section S3 for further details) of $B$ to $2B_0$ ($B_0 = 314$ MPa[45]) we achieve a good agreement of the shockfront trajectories $R_{SW}(\Delta t)$ with our data (cf. Fig. 5a and Supplementary Fig. 9a–c).

The numerical simulations yield spatial pressure distributions $p(R)$ which we compare to the experimentally determined profiles in Fig. 5b for the low $E_B$ and in Fig. 5c–e for the high $E_B$ simulations. Regarding the average pressure and not the functional form of the profile $p(R)$, we observe reasonable agreement for both energy ranges (see also $p_{peak}(R)$ in Supplementary Fig. 9d), only the average pressure for late $\Delta t \approx 15$ ns and high $E_B$ (Fig. 5e) lies significantly below the

experimental data. The line shapes of $p(R)$ agree well only at low $E_B$, even though also here the experimental curves show some distinct features not found in the simulated profile. More importantly, for high $E_B$, pronounced deviations appear. The experimental profiles $p(R)$ are more highly peaked or exhibit a higher slope, which is at intermediate and late $\Delta t$ not even correctly predicted in its sign. To show that this deviation is not a matter of our selection of events, we include a variety of different $p(R)$ distributions for individual cavitation bubbles within the corresponding energy range in Fig. 5c–e.

## Discussion

In summary, we have demonstrated that extreme states of cavitation bubbles can be probed by holographic imaging with nano-focused femtosecond FEL pulses, at high spatial and temporal resolution. Quantitative analysis of near-field diffraction patterns in the holographic regime gives access to physical conditions within the cavitation bubble, including the transition from early plasma state to a cavitation bubble, density profile, and shockwave pressure at different time delays $\Delta t$ after seeding. The technique offers the possibility of studying structural dynamics under different conditions (liquid parameters, external driving) in detail for a large ensemble of cavitation events. This makes it possible to study not only individual events, but simultaneously the entire ensemble, without uncontrolled ensemble averaging. In particular, all structural parameters can be sorted into bins of bubble radius, time after seeding, and/or bubble energy.

The shockwave shell bounded by the bubble radius $R_B$ and the outer shockwave radius $R_{SW}$ can be precisely quantified in terms of width and spatial density and pressure distribution as a function of time and bubble energy. Within this shell, the density and hence also the corresponding pressure is not constant, but

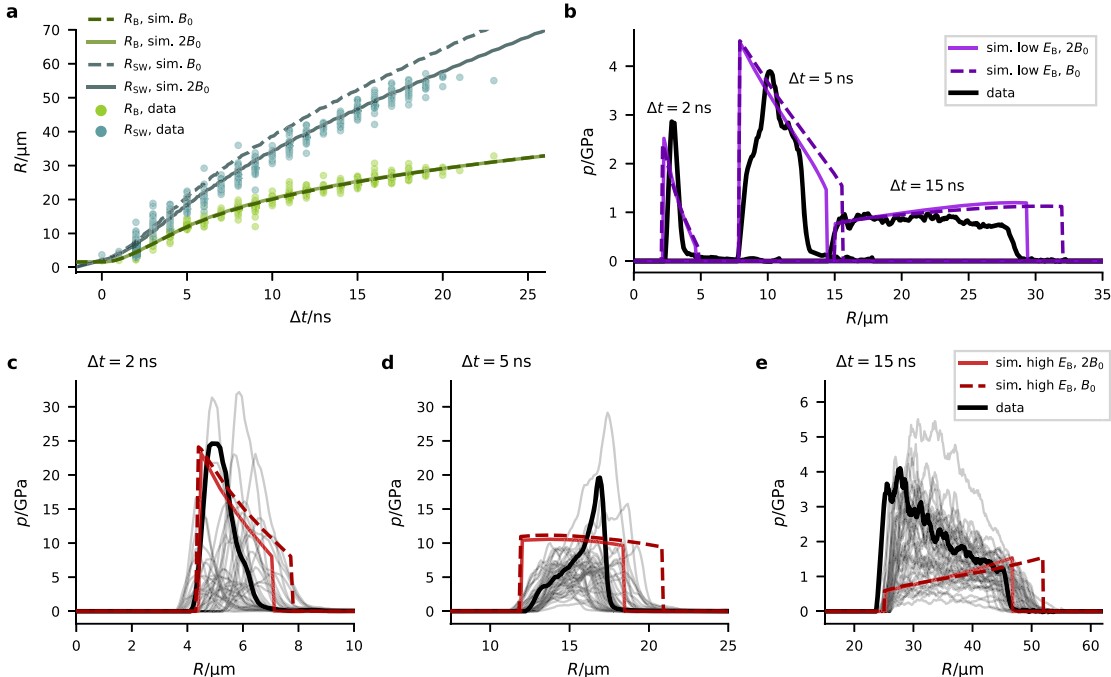

**Fig. 5 Simulations. a** Trajectories of the bubble wall radius $R_B$ and the shockfront radius $R_{SW}$ for the high $E_B$ simulation for both values of $B$. The energy range of $E_B$ for the experimental data shown here is between 111 and 130 μJ. The radius of maximal expansion of the simulations yields a bubble energy of 91 μJ. **b** comparison of the measured shockwave's pressure profile $p(R)$ with the simulated $p(R)$ (low $E_B$ simulation, $E_B \approx 20$–$33$ μJ) for three different time delays, again for both values of $B$. The time delay of the simulated profile was chosen such that it represents the experimental profile best. The exact time delays of the experimental data is $\Delta t = 2$ ns, 5 ns and 15 ns, and $\Delta t = 1.4$ ns, 6.5 ns and 13.3 ns for the simulations. The experimental pressure profiles are the same as in Fig. 3 with $E_B \approx 22 \pm 6$ μJ. **c–e** same as (**b**), but now for the high $E_B$ simulation. The three different time delays $\Delta t$ are indicated in the top left corner. The bold black curve shows the pressure profiles from Fig. 3 with $E_B \approx 119 \pm 4$ μJ. The gray curves are a selection of pressure profiles within the energy range shown in (**a**).

exhibits a peak, which quickly builds up with $\Delta t$ or correspondingly $R_B$, reaching a maximum $p_{max}$ at around $R_B \simeq 10$ μm, before it decays again more slowly with $R_B$. $p_{max}$ is a function of bubble energy and can exceed 20 GPa (Fig. 3a, b). The pressure profile as a function of $R$ is asymmetric, in particular for large $R_B$, where the maximum is near the inner interface and the pressure then decreases almost linearly to the equilibrium value (cf. Fig. 3c). Contrarily, at $R_B \simeq 10$ μm, i.e., when compression is highest in the shockwave, density and pressure accumulate at the outer interface (Fig. 3b). Note that the density profile extracted from the holograms is independent of assumptions regarding any equation-of-state, while the pressure profile is not. Here we have used the Tait equation as the simplest empirical model, but the density profile can of course also be analyzed with respect to different equations of state. The widths of both interfaces (gas-shockwave and shockwave-liquid) are also of interest. The profiles exhibit a smooth transition from compressed vapor to liquid with no sharp phase boundary, in contrast to the interface profile of equilibrium bubbles. Of course, the apparent width could also result from effects of non-spherical bubble shape, but this can—at least to some extent—be excluded for bubbles with lower energy (blue/magenta curves in Fig. 4b) and higher $R_B$ (green/yellow curves in Fig. 4c). Note that in these cases the phase profiles do not exceed the maximum vacuum/water phase shift (dashed lines), which is an indication for the sphericity of the bubbles.

The spatial density or pressure distribution close to the bubble nucleus can not be measured with optical or acoustic methods. Optical measurements could only determine a single pressure value at the shockfront from shockfront velocity measurements. For this reason we find significantly higher peak pressures, even for lower bubble energies, within the shockwave shell compared to optical shockfront observations[9]. Hydrophones for acoustic

measurements disturb the shock evolution when placed in too close proximity to the cavitation center. In addition, the hydrophones average over different radii, as the hydrophone dimensions are large, compared to the shockfront curvature at early times. For the first time we were able to measure the spatial shockwave pressure close to the cavitation center, and to compare it to numerical simulations. The comparison with numerical simulations showed a reasonable agreement with the overall peak pressure evolution (cf. Supplementary Fig. 9d). However, the functional form of the pressure profiles shows a pronounced discrepancy (cf. Fig. 5b–e). The deviations of the high $E_B$ simulations to the data is stronger than for the low $E_B$ case. In order to rule out that this discrepancy is an artifact of elliptical or conical shape deformations along the beam axis, which would not be correctly accounted for in the reconstruction, we have carried out analytical and numerical simulations, see section S2 of the Supplementary information. These show that for realistic deformation amplitudes, the density profile in the shockwave, if reconstructed under false shape assumption, would only be scaled but not altered in shape. At the same time, the orthogonally positioned optical camera helped to rule out events with multiple plasma cores and correspondingly stronger deformations. At the same time, we cannot exclude that already moderate deformations could lead to variations of the shockwave along the directions parallel to the the bubble surface. Also, the optical camera cannot resolve the early stages with potentially stronger asymmetry. However, by reducing the laser power to the sub-threshold regime of bubble seeding, the probability of strongly asymmetric events was significantly reduced. It is also important to note, that the higher order modes of bubble deformations are strongly damped, see section S2 of the Supplementary information. In future, the bubble shapes could be further controlled by observing

the cavitation bubbles perpendicular to the pump-laser beam axis. In such a geometry, a possible variation of the shockwave density in different directions from the bubble center could be probed, which would be an interesting effect in itself to be targeted in a follow-up experiment. In that case one would need to use a 2d-phase retrieval approach (e.g., AP, cf. Supplementary Fig. 1d, e) and the Abel transform for cylinder symmetry. A comparison with numerical simulations carried out with full spatial dimensionality (3d)[48] could also shed light on how crucial the exact shape of the bubble influences the spatial pressure distribution of the shockwave.

Importantly, however, realistic shape distortions can not explain the inversion of the pressure slope between simulation and data. We therefore must attribute the main discrepancy to the model assumptions. Notably, the Gilmore model approximates the Mach number up to the first order. Cavitation bubbles of higher energy and velocity are therefore less accurately described by the model. With the capability to probe the density profile directly by holographic X-ray imaging, new theoretical approaches beyond the current models are now timely and promising, since the predictions could be put under direct experimental validation. Correspondingly, the pressure profiles presented here could guide novel theoretic work.

The direct accessibility of density profiles also motivates evaluation and development of more advanced models in the future. In such efforts, the equations of state of water should be put into question. Incorporation of more details of optical breakdown, plasma growth[49], phase transition and heat exchange[50] could be addressed as well as higher-order liquid compression terms in spherical bubble models, as well as non-spherical laser plasma shapes, which can be treated by 3d fluid dynamics simulations [26,48].

The methodology presented here can also be applied to more complex environments, such as cavitation interaction with a wall or interface. More generally the method can be extended to different sample systems, from driven complex fluids, to plasmas and warm dense matter. The spatial resolution was limited to about ≲500 nm, which can be attributed to the dispersive focusing effects of the SASE pulses by the CRL. By either increasing monochromaticity with, e.g., seeded SASE pulses or by the use of achromatic nanofocusing optics with high numerical aperture, the resolution could be scaled up by more than an order of magnitude, see section S1 of the Supplementary information for a detailed discussion of resolution and scalability. While we have focused here on the bubble trajectory after seeding in a regime where a nanosecond-pump laser was sufficient, picosecond or femtosecond pump pulses would allow to investigate the ultra-fast time scales of optical breakdown in water, plasma generation and the nascent state of bubble generation.

With a future extension of the presented method, vital questions on the bubble collapse, associated with single-bubble sonoluminescence, could be answered. To this end, the collapse of the bubbles needs to be predictable with nanosecond accuracy. This could be achieved by trapping the cavitation bubbles in a stationary ultrasonic field[1,2], synchronizing the bubble trajectory to the ultrasound (see Supplementary Section S4 for more details). The exact radii of collapsing bubbles are not known experimentally, but are smaller than 1 μm in diameter and can therefore not be resolved with visible light. Numerical models[18–21] predict an inhomogeneous, fast evolving distribution of pressure, density and temperature for the bubble collapse, with converging compression or shockwaves, demixing, chemical reactions and the formation of a nanoscopic thin-plasma core[3,8,18] which is supposed to be the source of cavitation luminescence. With the presented methodology, direct experimental validation of this scenario is now within reach.

## Methods

### Experimental design

*X-ray optics.* The experiment was performed at the MID (Materials Imaging and Dynamics) instrument[33,34] at the European X-ray Free-Electron Laser[35] in Schenefeld, Germany. The FEL was operated at 14 GeV electron energy and an undulator line delivered ultra-fast (100 fs or less) X-ray pulses with 14 keV photon energy, 10 Hz repetition rate in single-bunch mode and 600(300)μJ average pulse energy or about $3(2) \times 10^{11}$ photons per pulse. The X-rays were focused by a stack of 50 nano-CRLs, aberration corrected by a custom-made phase plate[51], with a focal length of 298 mm and a numerical aperture of $4.3 \times 10^{-4}$. Prior to the nano-CRLs, the *CRL-1* system of the MID instrument[33] was used to prefocus the X-rays. The prefocus was chosen such that the beam size at the nano-CRLs overilluminated the nano-CRLs' aperture. The X-ray focus to sample distance was $z_{01} = 144$ mm and focus to detector distance $z_{02} = 9578$ mm. The X-ray detector was a five mega pixel sCMOS camera (Andor Zyla 5.5, Oxford Instruments, Abingdon, United Kingdom) with a fiber-coupled scintillator (LuAg:Ce, thickness 20 μm) converting X-rays to optical photons with a pixel size of 6.5 μm. The cone-beam geometry led to 66.5× magnification and 98 nm effective pixel size in the sample plane. The Fresnel number, describing the wave-optical properties of the imaging system, was $F = 7.6 \times 10^{-4}$. In the sample cuvette, the X-rays passed two quartz-glass windows with 150 μm thickness and about 5 mm of water.

*Laser optics.* We used a Litron Lasers Nano L 200-10 (Litron Lasers, Rugby, United Kingdom) laser system with 1064 nm wavelength, 6 ns pulse duration and 200 mJ maximum pulse energy, which was reduced to 24 mJ by an internal attenuator. The beam was expanded to increase the numerical aperture to 0.2, with a focal length of 50 mm. A flat mirror with a through-hole allowed co-linear alignment of the laser and X-ray beam. The focal spot size is expected to exceed the diffraction limited FWHM of 1.7 μm, since in addition to spherical abberations of the lens, the through-hole on the last mirror introduced aberrations to the wavefront and a fine adjustment of the laser focus position was used to match the laser and X-ray focus after focal alignment of the laser. The seeding rate of the cavitation events was about 23% with a Root-Mean-Square variation of 3%. Multi-bubble events have been observed for about 30% of the cavitation events. The radius of maximum expansion of the cavitation bubbles was typically in the range of 500–700 μm, with lifetimes of 100–150 μs. A detailed analysis of these properties is published separately in ref. [37].

*Optical high-speed measurements.* Observation of the individual cavitation events with the optical high-speed camera (Photron Fastcam SA5, Photron, Tokyo, Japan) allows to capture the full bubble dynamics, including plasma breakdown, expansion, first collapse and bubble rebound from the side. Images where recorded with background illumination with a continuous halogen light source (LS-M352, Sumita, Japan) using a long-distance microscope (K2 Distamax, Infinity, USA). Incoming light is refracted by the cavitation bubble, creating a shadow in the bright-field image. From the optical imaging we deduce the number and shape of plasma luminescence spots and follow the full bubble motion, including the measurement of its maximum expansion radius $R_{max}$. Due to limitations in the data download speed from the optical camera's internal memory, optical measurements are conducted for approximately half of all runs. Additional information and exemplary high-speed recordings can be found in Supplementary Fig. 1.

*Timing equipment.* In order to process each cavitation event individually, precise timing, as well as the ability to relate each data source to one unique cavitation bubble is necessary. The FEL provided a unique train ID for each X-ray pulse, which was stored along with the signals acquired by the MID instrument. However, custom equipment, not fully integrated to the FEL's data acquisition system (DAQ), was necessary for this experiment. To this end, an AND gate was used to synchronize data sources not integrated in the FEL's DAQ with the FEL's unique train IDs. The AND gate provided a centralised first pulse, so that pump laser, high-speed camera, and the data recording of the microphone started simultaneously. The output of the AND gate was fed to the FEL's DAQ, so that this first pulse could be attributed to the unique train ID of one X-ray pulse.

For the precise timing, we used a pair of low jitter delay generators (DG535, Stanford Research Systems) controlling the delays of the lasers flash lamp, Pockels cells, and the high-speed optical camera to the FEL's master trigger. The delay between the flash lamp and Pockels cells was kept constant at 160 μs for maximal laser output. The signals of the microphone, the FEL's master trigger, the output of the laser Pockels cells and the shutter output of the high-speed optical camera were digitised by an USB oscilloscope (PicoScope 6402C, Pico Technology, St Neots, United Kingdom). Further details on the timing setup, including cabling schemes and all electronic components are published in[37].

## Data analysis

*Phase retrieval: radially fitted phase.* Propagation of radially symmetric wavefields: The two dimensional (2d) Fourier transform $\mathcal{F}$ of a 2d signal $f(x, y)$ with radial symmetry $f(x, y) = f(r \cos \theta, r \sin \theta)$ is related to the zeroth-order Hankel transform $\mathcal{H}_0$ as[52]

$$\mathcal{H}_0[g](\nu) = \frac{1}{2\pi} \mathcal{F}[f](\nu \cos \theta, \nu \sin \theta). \quad (2)$$

As the zeroth-order Hankel transform is self-inverse, the 2d Fourier transform of a radially symmetric signal is (up to prefactors) self-inverse as well.

The 2d Fresnel propagator is written as[53]

$$\psi(x, y, z = \Delta) \approx \exp(ik\Delta) \cdot \mathcal{F}^{-1}\left[\exp\left(\frac{-i\Delta(\nu_x^2 + \nu_y^2)}{2k}\right) \mathcal{F}[\psi(x, y, z = 0)]\right], \quad (3)$$

with $\psi(x, y, z)$ the wavefield at position $(x, y, z)$, where $z$ is the direction of propagation and $\Delta$ the propagation distance. $\mathcal{F}$ is the 2d Fourier transform in a plane perpendicular to the propagation distance and $(\nu_x, \nu_y)$ the Fourier coordinates. $\mathcal{F}^{-1}$ is the inverse Fourier transform, respectively. Note that the Fresnel kernel is radially symmetric, as it only depends on $\nu_x^2 + \nu_y^2 =: \nu_\perp^2$. This implies that the propagated wavefield $\psi(x, y, z = \Delta)$ of a radially symmetric wavefield $\psi(x, y, z = 0) = \psi(r_\perp, z = 0)$ at $z = 0$ has radial symmetry as well and thus only depends on $(r_\perp, z)$. Using Eq. (2) we can write the propagated wavefield as

$$\psi(x, y, z = \Delta) = \psi(r_\perp, z = \Delta) = \exp(ik\Delta) \cdot \mathcal{H}_0\left[\exp\left(\frac{-i\Delta \nu_\perp^2}{2k}\right) \mathcal{H}_0[\psi(r_\perp, z = 0)]\right]. \quad (4)$$

The discrete Hankel transform can be written as a matrix multiplication of an $N \times N$ matrix $H_0$ with an $N \times 1$ vector representing a discretization of a function $f$[54]. With a discrete kernel of the Fresnel propagation $D_\Delta$, this gives a fast and efficient Fresnel-type propagator in radial symmetric coordinates

$$\psi(r_j, z = \Delta) = \exp(ik\Delta) H_0 D_\Delta H_0 \psi(r_j, z = 0). \quad (5)$$

The propagation matrix $P_\Delta = H_0 D_\Delta H_0$ has to be calculated only once and can be used for propagation of different wavefields $\psi$, so that the Fresnel propagation reduces to the matrix multiplication of $P_\Delta$ with a wavefield $\psi$.

Radially Fitted Phase: The phase retrieval approach Radially Fitted Phase makes use of the radial symmetry of the cavitation bubbles and is formulated as an optimization problem, searching for the object's phase $\overline{\phi}(R)$ minimizing the $\ell^2$-distance of the calculated radial intensity $I(\overline{\phi}(R))$, when propagating $\overline{\phi}$ numerically to the detector, to the measured radial intensity $I_{\text{meas}}$, i.e., $||I_{\text{meas}}(R) - I(\overline{\phi}(R))||^2$. For the calculation of $I(\overline{\phi})$, the object's exit field is calculated in a first step, using a constant $\beta/\delta$-ratio $\kappa$. The assumption of constant $\kappa$ is perfectly satisfied for the cavitation bubbles containing water and water vapor at different pressures only. In a second step, the object's exit wavefield $\psi_{\text{obj}}(R) = \exp[(i + \kappa)\overline{\phi}(R)]$ is propagated to the detector, using the matrix approach from Eq. (5). The minimization of the $\ell^2$-norm is done by the BFGS algorithm[55], a quasi-Newton method by Broyden, Fletcher, Goldfarb, and Shanno implemented in the *minimize* function of *SciPy*'s *optimization* submodule (version 1.4.1)[56]. The method can be easily extended to be regularized by further penalty terms, such as total variation (TV) norm or Tikhonov regularization. For the data shown in this work the algorithm was stable without further regularization.

Regularized inverse Abel transform: The phase retrieval gives access to the projected phase of the cavitation bubbles, but to access physical quantities, the 3d phase of the cavitation bubble is indispensable. The inverse Abel transform[57] gives a fast and efficient way to calculate the 3d phase from its projection as a linear map, with the assumption of spherical symmetry. However, the reconstruction of the central voxels of the 3d-phase distribution is strongly affected by noise, as the number of voxels per shell with radius $R$ decreases quadratically. To stabilize the inverse Abel transform against noise, we regularized the inner voxels with an $\ell^1$-norm total variation penalty term up to a radius $R_{\text{TV}}$ which is 60 % of the radius $R_\text{B}$ where the bubble transitions into the shockwave. This regularizes about 36 % of all voxels of the gaseous bubble and even less of the whole volume, including the shockwave. The optimization uses *SciPy*'s *minimize* function as described in the paragraph above.

*Acoustic signal.* The acoustic signal is detected by a piezoelectric microphone, glued to the outside of one of the cuvette walls. The acoustic waves emitted by the optical breakdown and the collapse are recorded by the USB oscilloscope (PicoScope 6402C, Pico Technology, St Neots, United Kingdom), with a sampling rate of 38.4 ns. At the position of the microphone, in a distance of ~15 mm from the breakdown position, the shock and sound waves are dispersed. Noise originating from reflections from the cuvette walls, and further scattering from impurities and satellite bubbles are present. The lifetime $\tau$ is obtained as the time interval between the first two strongest peaks of the convolved microphone intensity (rectangular kernel with a width of 38.4 μs). For random samples, we verified that these two peaks correspond to the breakdown and first collapse of the cavitation events.

*Classification of cavitation events.* Individual cavitation events are classified in terms of the mechanical bubble energy $E_\text{B}$ that is deposited by the IR laser pulse. This value can be accessed from the maximum expansion radius of the cavitation bubble $R_{\text{max}}$, related by[46,58]

$$E_\text{B} = \frac{4}{3}\pi(p_0 - p_\nu)R_{\text{max}}^3. \quad (6)$$

Here $p_0 = 100$ kPa is the ambient hydrostatic pressure and $p_\nu = 2.34$ kPa the vapor pressure at ambient temperature of $T_0 = 20$ °C[59]. Since direct measurement of $R_{\text{max}}$ by the high-speed optical camera is available only for about half of all events, we extrapolated the relation between the lifetime $\tau$, which is obtained from the acoustic signal, and $R_{\text{max}}$. For a spherical collapse this relation is given by[46,60]

$$\tau = 2 \cdot 0.915 \, R_{\text{max}} \sqrt{\frac{\rho_0}{p_0 - p_\nu}}, \quad (7)$$

with $\rho_0 \simeq 1$ g cm$^{-3}$ the equilibrium water density. Note that the lifetime $\tau$ is assumed to be twice the collapse time. We observe a linear relation of $R_{\text{max}} = m \cdot \tau + b$ with $m = 4.45(3)$ m s$^{-1}$ and $b = 84(3)$ μm. Hence the measured collapse time is prolonged by a factor of 1.22 with respect to the spherical case, given by the Rayleigh-Plesset model. In part, this is expected to be induced by boundary interaction of the cavitation bubble with the entrance window. The offset $b$ can not fully be attributed to the initial size of the breakdown plasma. Further details are published in[37].

## Numerical modeling of cavitation and shockwave dynamics

*Bubble dynamics.* The dynamics of the early bubble growth was simulated with a Gilmore-Akulichev model[32] in combination with shockwave propagation based on the Kirkwood-Bethe hypothesis[47]. This model is usually used for simulations including acoustic radiation as it incorporates both liquid compressibility as well as a pressure-dependent sound velocity[61]. We implemented a time-dependent absorption of the laser pulse energy into the Gilmore model as was previously used in[9].

The calculation is based on two steps—the first step is the simulation of the bubble boundary motion via the solution of the following system of differential equations for the position $R$ and velocity $U$ of the bubble wall:

$$\dot{R} = U \quad (8)$$

$$\dot{U} = \left[-\frac{3}{2}\left(1 - \frac{U}{3C}\right)U^2 + \left(1 + \frac{U}{C}\right)H + \frac{U}{C}\left(1 - \frac{U}{C}\right)R\frac{dH}{dR}\right] \cdot \left[R\left(1 - \frac{U}{C}\right)\right]^{-1}, \quad (9)$$

with the pressure dependent sound velocity $C$, the enthalpy $H$ and pressure $P$ at the bubble wall, given by

$$C = c_0\left(\frac{P + B}{p_0 + B}\right)^{\frac{n-1}{2n}}, \quad (10)$$

$$H = \frac{n(p_0 + B)}{\rho_0(n - 1)}\left[\left(\frac{P + B}{p_0 + B}\right)^{\frac{n-1}{n}} - 1\right], \quad (11)$$

$$P = \left(p_0 + \frac{2\sigma}{R_n}\right)\left(\frac{R_n}{R}\right)^{3\kappa} - \frac{2\sigma}{R} - \frac{4\eta U}{R}. \quad (12)$$

Here, $c_0 = 1483$ m s$^{-1}$ is the sound velocity in water at normal pressure $p_0 = 100$ kPa[59], $n = 7$ and $B = 314$ MPa are empirical parameters of the Tait equation of state[45], $\rho_0 = 998$ kg m$^{-3}$ is the density of water, $\sigma = 72.538$ mN m$^{-1}$ the surface tension at the water-vapor interface, $\kappa = 4/3$ the polytropic exponent and $\eta = 1.046$ mPa s the dynamic viscosity of water at room temperature[9]. The bubble interior is modeled as an ideal gas. The laser pulse is assumed to be Gaussian-shaped, and incorporated by the time-dependent rest radius

$$R_n(t) = R_{nb}\left[0.5\left(1 + \text{erf}\left(\frac{t - t_\text{a}}{\sigma_l\sqrt{2}}\right)\right)\right]^{\frac{1}{3}}. \quad (13)$$

The increase of vapor volume of a sphere with radius $R_n(t)$ is proportional to the deposited laser energy. In this way, $R_n$ expands during the presence of the laser pulse and is constant afterwards, driving the rapid expansion of the cavitation bubble. We used the error function erf($t$) with the width $\sigma_l$ to compute the energy deposition of the Gaussian shaped pulse with a FWHM width of $\tau_l = 2\sigma_l\sqrt{2\ln(2)}$. The effective initial bubble radius $R_{na} = R_n(t = 0)$ is varied via $t_\text{a}$. We typically choose $R_{na} \approx 1$ μm, being significantly smaller than the radius of the initial plasma spark observed by the optical camera.

*Shockwave propagation.* The second step of the simulation is the calculation of the pressure profile for radii $r$ beyond the bubble wall radius $R$ ($r > R$), via shockwave propagation. To this end, we compute the trajectories of the characteristics using each state of the bubble wall trajectory as initial conditions for the propagation of the invariant quantity $G = r(h + u^2/2) = R(H + U^2/2)$ by solving the following

system of differential equations[47,60,62]:

$$\dot{r} = u + c \tag{14}$$

$$\dot{u} = \frac{1}{c-u}\left( (u+c)\frac{G}{r^2} - \frac{2c^2 u}{r} \right) \tag{15}$$

$$\dot{p} = \frac{\rho_0}{r(c-u)}\left( \frac{p+B}{p_0+B} \right)^{\frac{1}{n}}\left( 2c^2 u^2 - \frac{c^2+uc}{r}G \right) \tag{16}$$

Here, $r$ is the position, $u$ the velocity and $p$ the pressure of the characteristic. Further parameters, such as the pressure-dependent sound velocity $c$, are given in the previous paragraph.

Pressure profiles are found as plane-intersections of constant $t = \Delta t$ in the $p(r,t)$-space spanned by all characteristics. At the shock front, a discontinuity is present, indicated by ambiguous distributions $u(r)$ and $p(r)$. As prescribed by the conservation laws of mass-, momentum- and energy-flux through the discontinuity, the position of the shock front $r_s$ is determined to be at the position, where the area below and above the ambiguous part of the respective $u(r)$ curves are equal[9,63]. For $\Delta t$ where a shock has not yet formed, the front of the pressure profile was determined as the width where the pressure surrounding the bubble drops to $1/e^2$ of its peak pressure. The model assumes a constant gas pressure $p(r < R) = P$ inside the cavity, and equilibrium pressure $p(r > r_s) = p_0$ beyond the shock front.

We optimize the parameters $R_{na}$, $R_{nb}$, $\tau_l$, and $t_0$, so that the simulated trajectories of the bubble wall $R_B(\Delta t)$ and shock front position $R_{SW}(\Delta t)$ fit with the experimentally determined values from X-ray imaging, as well as with the optical and acoustic measurements. The time shift $t_0$ is used to determine the arrival of the seeding laser with respect to the FEL pulse.

Alternatively, we are able to compare directly the simulated pressure profiles $p(r > R)$ to the data obtained by X-ray holography.

## Data availability

The data that support the findings of this study are available from the corresponding author upon reasonable request and after 2022-06-10 through European XFEL services under[64].

## Code availability

The algorithms used to reproduce the findings of this study are described in detail within the manuscript and the Methods. The code is available from the corresponding author upon reasonable request.

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

## Acknowledgements
We thank Peter Luley and Jan Goemann for technical help. The authors thank the European XFEL in Schenefeld, Germany for granting beam time for this project. The engineering team and technicians at the MID instrument are gratefully acknowledged for help in setting up the experiment. This research was supported in part through the Maxwell computational resources operated by DESY. Open Access funding enabled and organized by Projekt DEAL.

## Author contributions
T.S., J. Hagemann authored the proposal of the XFEL experiment, with help from M.O., T.K. and R.M.. M.V., H.P.H., J. Hagemann, J.M.R. and M.O. prepared the experiment from the user side, with support/advice from T.S., R.M. and T.K.. M.S., J.M., J. Hallmann, U.B., C.K., A.Z., W.L., R. Shayduk, R. Schaffer, led by A.M. prepared and contributed instrumentation and integration work by M.O., M.V. and H.P.H.. F.S., A.S. and C.G.S provided the nano-focus optic. M.V., H.P.H., J. Hagemann, J.M.R., M.O., A.S. and T.S. performed the XFEL experiment with M.S., J.M., J. Hallmann, U.B., C.K., A.Z., W.L., R. Shayduk, R. Schaffer and A.M., M.V., H.P.H., M.O. and J. Hagemann analyzed data with input from T.S. and R.M. The manuscript was mainly written by M.V. and T.S. with substantial input from co-authors. All authors read and approved the manuscript.

## Funding
M.V. and T.S. are members of the Max Planck School of Photonics supported by BMBF, Max Planck Society, and Fraunhofer Society, which has also funded consumables and instrumentation for the cavitation and optical setup. J. Hagemann and C.G.S. have been funded by the Helmholtz Imaging Platform (HIP), a platform of the Helmholtz Incubator on Information and Data Science. We acknowledge funding of the nanofocusing setup at MID by BMBF project 05K13OD2 Erzeugung und Charakterisierung von nanofokussierten XFEL-Pulsen zur Abbildung ultraschneller Prozesse in Materie and for the IR-laser by BMBF project 05K16RF2 Mikroskopische Flüssigkeitsstrahlen zur Untersuchung der Dynamik und Kinetik struktureller Nichtgleichgewichtsphasenübergänge am Europäischen Freie-Elektronen-Laser.

## Competing interests
The authors declare no competing interests.

## Additional information

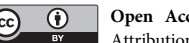

