## [Peer Review File · Nature Communications]

Editorial Note: Figures provided by Reviewer 2 in the second round in this Peer Review File have been redacted to remove third-party material where no permission to publish could be obtained.

Reviewers' Comments:

Reviewer #1:

Remarks to the Author:

Please see the attached report.

Report on the Paper:
**Pump-probe X-ray holographic imaging of laser-induced cavitation
bubbles with femto-second FEL pulses**
by M. Vassholz *et al.*

What the paper is about: The paper deals with a novel experimental technique to investigate the dynamics of laser-induced cavitation bubbles. A pulsed infra red (1064 nm) laser (pump) is used to repeatedly generate cavitation bubbles inside a water-filled cuvette (6 ns pulse duration and 24 mJ pulse energy). Purpose of the paper is demonstrating near-field holographic imaging of cavitation bubbles with single X-ray free-electron laser (XFEL) ultra short pulses (order of 100 fs) (probe) to investigate the extreme states of bubble generation and collapse. An X-ray camera is used to record the holograms and acquire order 3000 sample cavitation events. A high speed optical camera allows to observe sideways the process simultaneously with X-ray acquisition while acoustic signals associated with shock wave emissions were detected by a piezo-ceramic microphone at the cuvette wall. The experiments were performed at the MID (materials and imaging dynamics) instrument of the European XFEL which provides ultra-short pulses at a repetition rate of 10 Hz.

Novelty and groundbreaking nature of the results: The use of X-ray holography to investigate single bubble cavitation events triggered by optical breakdown in water is new and certainly interesting. The technique requires the availability of an advanced X-ray source, like the free-electron laser used by the authors. On top of that, the paper describes two significant technical advancements related to the analysis of the holograms generated by interference of the reference beam with the signal scattered by the bubble, namely: i) A correction strategy is implemented to take into account the fluctuations inherent to the beam source. Dynamic flat field correction is applied to the holograms to correct for beam fluctuations, following Van Nieuwenhove *et al.*, Optics Express 2015 concerning synchrotron radiation, Ref. 20, and Ref. 21, to be published by the authors – and unavailable to the referee – concerning XFEL beams. The idea is to use a Principal Component Analysis of XFEL flat (i.e. in the absence of the scattering bubble) fields and correct each hologram by the weighted combination of eigen fields than minimizes the total variation of the corrected intensity. ii) Under the assumption of bubble spherical symmetry, the phase of the scattered signal is retrieved, or more properly reconstructed, by minimizing (the ℓ^2 -norm of) the difference between measured (corrected and radially averaged) hologram intensity and the intensity that results from numerically propagating (through a Fresnel propagator) the beam subject to the object-induced phase shifts. As a result of the optimization, the phase field is determined. Under the assumption of spherical symmetry the Abel transform is used to deconvolve the phase shift and determine the contribution of the inner structure of bubble and emitted shock wave. From the radial phase the authors have access to the fluid density radial distribution and from the density they estimate the pressure using a presumed equation of state (here Tait's EOS).

Concerning cavitation bubble dynamics, the results discussed in the paper are mostly illustrative of the method potential. Graphs are provided to demonstrate the radial distribution of phase across the bubble and the related pressure/density estimates. The data are analyzed based on the median radial phase shift in a group of acquisitions in a certain range of bubble mechanical energy as inferred from the bubble life time determined through the acoustic signal measured by the hydrophone. Standard formulae based on a classical Rayleigh type description of bubble dynamics which relate bubble life-time and maximum bubble diameter to the energy are used to the purpose. As a general comment no substantial novelty seems to emerge concerning our understanding of cavitation and bubble dynamics.

Overall paper quality: The paper is well written and clear, and I did not spot typos or misprints, which in my reviewer experience is sort of unusual. Figures and related caption are illustrative of the reported work. The relevant technical aspects are well explained and the results appear to be physically sound. Although I do not deem myself a specialist of the field, significant elements of novelty are discussed concerning X-ray holographic data analysis. The proposed technique may potentially allow to obtain significant new data on the cavitation process, at least in the context of single bubble events. The authors claim that their configuration can be (easily?) extended to deal with near wall cavitation, which is of sure interest for many applicative fields, and to address the extreme states of the system related to bubble collapse and plasma dynamics.

Issues and comments:

- It is well known that single laser-induced cavitation bubbles undergo a characteristic sequence of phases: an initial phase of optical breakdown and plasma formation, a second phase of bubble expansion and a final phase of bubble collapse which, depending on degree of symmetry and bubble energy, may lead to successive rebounds. At collapse, a shock wave is launched in the surrounding liquid environment. This is the shockwave related to cavitation dynamics. A shockwave is also associated with the breakdown phase. From the discussion and from figure 3 and 4.c the authors are seemingly dealing with the breakdown shockwave. The impression is that the text would gain in clarity if an explicit comment is added to explain the issue.
- Concerning laser induced bubbles, a crucial point is the symmetry achieved in the system. The latter depends on plasma shape and fragmentation, elongation along optical axis and asymmetry with respect to the pump laser focal point. The authors should qualify their data by showing and discussing the plasma images captured from the side-view optical camera.

- The authors introduce an ellipticity factor in the formula used to convert radial phase into radial density profile. It would be nice to correlate this effective parameter, used to prevent the reconstructed density from decreasing below vapor pressure, with the shape of the mature bubble. In fact, any initial lack of sphericity is amplified during the bubble collapse phase making it easy to quantify with the side camera.
- From the point of view of cavitation, a central point to discuss is the bubble collapse, entailing, if simulations are not mistaken, a strongly focused compression wave and, possibly, transition of the vapor to supercritical conditions. In principle, as claimed by the authors, the proposed set-up should enable to follow this phase. Is there a reason why the authors do not show data for the collapse? Did they encounter substantial difficulties in dealing with that? In the present referee view point, adding even partial data on this aspect would crucially demonstrate the strength of the approach and substantially enhance the value of the paper.
- Do the authors believe their technique could be extended to axisymmetric bubbles? Ideally, the bubbles are more likely to be axisymmetric along the optical axis of the pump laser beam rather than spherical. I imagine that, by taking holograms from the side, more or less the same analysis technique should work except that a cylindrical rather than spherical Abel transform would be needed to reconstruct the bubble interior. Unless I am wrong, the authors should explicitly comment on this important issue (see also the last item of the present list).
- The study of extreme states associated with optical breakdown, plasma formation, bubble collapse/rebound and shock wave dynamics are presented as perspective applications, as suggested by the sentence *An increase in the frame rate of detectors used for ultra-fast holographic imaging to kHz would enable ..* . In order to resolve the implied fast dynamics, an extremely fast sampling rate is mandatory. The author should illustrate more in depth the available detector technology and related limitations and discuss how far is the state of the art from achieving the required resolution.
- The Fresnel number of the optical system is given as $F = 7.6 \times 10^{-4}$ which, according to classical definitions, should correspond to the far-field, Fraunhofer regime. It is then not clear why is the approach referred to as *near-field* holographic imaging. Am I missing something?
- The proposed analysis is deeply routed on the assumption of spherical symmetry. On the contrary near wall cavitation bubble dynamics is never spherically symmetric, implying the formation of high speed jets toward or away from

the boundary, for solid walls and free surfaces, respectively. In order to avoid conjectural sentences and for the interest the issue may have, can the authors explain how they plan to extend their approach to such much more complicated geometrical conditions?

Recommendation: In conclusion, I read the paper with sincere interest and I understood the reported novelty as being related more to the field of X-ray holography rather than to bubble dynamics and cavitation. I believe that the authors should make an additional step toward actually demonstrating the potential of their new holographic technique for the specific field of single cavitation bubble dynamics by addressing the comments reported in the list above in view of the eventual publication of their paper in Nature Communications.

Reviewer #2:

Remarks to the Author:

The authors present a time-resolved imaging study of laser-induced cavitation bubbles in water at the micrometer and nanosecond time scales. The setup is very elaborate, the data analysis is detailed and well explained. However, I would recommend to reject the publication of this manuscript in Nature Communications due to insufficient justification of general interest.

The authors suggest that their paper presents a new imaging method for shock waves with significant advantage over previous visualization attempts in spatial and temporal resolutions(1), and that their method delivers new insights into the evolution of electron density in shock fronts, which can be used to improve and calibrate existing theoretical models (2). In my opinion, at least one point (1) or (2) have to be fulfilled for publication in Nat. Comm, but the authors don't present enough evidence to support this impression.

Concerning point 1:

1a. Imaging of shock fronts with XFEL using holography has been successfully demonstrated in diamond and other materials in Schropp, Andreas, et al. "Imaging shock waves in diamond with both high temporal and spatial resolution at an XFEL." Scientific reports 5 (2015): 11089. This study is not, but should be cited in this manuscript. The imaging setup, and the temporal and spatial resolutions were very similar to the ones presented in the current study. Some variations in image analysis is the most prominent difference at the first glance.

1b. Also, it is true that the spatial resolution of near field X-ray holography is not limited by the detector pixel size in contrast to X-ray microscopy. However, it is not an accident that both methods achieve similar spatial resolutions. In most imaging experiments the resolution limit is determined by the brightness of the image features which carry the high spatial resolution information. This factor is mostly determined by the FEL pulse energy, which is not really scalable (in contrast to the pixel angular size). This is why it will be hard to image sub-micrometer bubbles with tens of nanometers resolution using the presented method. As such, both methods face similar limitations and near-field X-ray holography does not present a significant scalable improvement.

1c. Even a greater problem of the manuscript is that it is not well explained why the presented study can deliver results which differ greatly from high-resolution and high speed optical imaging. Especially for medium-sized bubbles in early stages of the expansion multi-scattering effects seem not to be very pronounced. The experimental work cited in the publication is mostly over 20 years old. For example, newer studies such as Charee, Wisan, and Viboon Tangwarodomnukun. "Dynamic features of bubble induced by a nanosecond pulse laser in still and flowing water." Optics & Laser Technology 100 (2018): 230-243. nicely visualize shock formation at the boundary of bubbles. This study, which relies on a much cheaper direct imaging method, exhibit a similar clarity of detail without the artifacts of inline X-ray holography such as ringing and the need of reconstruction algorithms.

Concerning point (2): In the introduction, the authors claim that high resolution X-ray imaging will help to understand and better simulate the formation of bubbles and electron density of shock fronts. However, in the discussion section there is no mentioning if the results are a surprise to or align well with theoretical predictions. Overall, it seems that the process the authors observe is very well understood. If this is not the case, the authors have to stress this point more vigorously.

In summary, I would suggest a publication in a more specific journal.

NCOMMS-20-28821-T: Response to Reviews & Summary of Changes

Reviewer 1:

What the paper is about: The paper deals with a novel experimental technique to investigate the dynamics of laser-induced cavitation bubbles. A pulsed infra red (1064nm) laser (pump) is used to repeatedly generate cavitation bubbles inside a water-filled cuvette (6 ns pulse duration and 24 mJ pulse energy)...

Reply: We thank the reviewer for the very careful reading, precise summary, positive assessment and for helpful suggestions in view of further improvements, which we have all followed, as summarized point-by-point below. In fact, we find the review to be exceptionally precise and comprehensive in understanding of the work. We just would like to strengthen one perspective which we may not have sufficiently stressed, namely that the methodology developed opens up the possibility to image a wide class of phenomena beyond cavitation, from strongly driven processes in water and liquids to dynamics of warm dense matter. This said, we here focus entirely on cavitation, and the main scope of the revision was to extend the analysis and to show that new insights can be derived for this application, notably by analysis of the shockwave density, and additional comparison to cavitation models.

Novelty and groundbreaking nature of the results: The use of X-ray holography to investigate single bubble cavitation events triggered by optical breakdown in water is new and certainly interesting... On top of that, the paper describes two significant technical advancements related to the analysis of the holograms generated by interference of the reference beam with the signal scattered by the bubble...

Reply: We are pleased that the reviewer considers our work to be new, certainly interesting, and that he/she acknowledges significant technical advancements.

From the radial phase the authors have access to the fluid density radial distribution and from the density they estimate the pressure using a presumed equation of state (here Tait's EOS). Concerning cavitation bubble dynamics, the results discussed in the paper are mostly illustrative of the method potential. Graphs are provided to demonstrate the radial distribution of phase across the bubble and the related pressure/density estimates. The data are analyzed based on the median radial phase shift in a group of acquisitions in a certain range of bubble mechanical energy as inferred from the bubble life time determined through the acoustic signal measured by the hydrophone. Standard formulae based on a classical Rayleigh type description of bubble dynamics which relate bubble life-time and maximum bubble diameter to the energy are used to the purpose. As a general comment no substantial novelty seems to emerge concerning our understanding of cavitation and bubble dynamics.

Reply: In response to this important point, we have now significantly extended the analysis based on computational fluid dynamics. Beyond the basic discrete bubble models according to Rayleigh, and Rayleigh-Plesset, we have included a full solution of the common Gilmore-Akulichev model, which includes a non-constant external pressure and the water compressibility. Fitting this model to the trajectories of the bubble wall radius and shockfront radius from our data (regrouped for bubble energy), we obtain spatial pressure distributions from a simulated shockfront propagation. We can then compare the measured spatial pressure distributions to the simulated profiles, which are obtained independently from the model. This has, in fact, never been the case before, since the fast visible light microscopy can only exploit a single pressure value at the shockfront from the bubble and shockfront dynamics $R(t)$. It is

important to note that in the Gilmore model, many assumptions are made ad hoc, in particular on the initial conditions, and that it also assumes a homogeneous (locally equilibrated) gas phase with a sharp phase boundary to the fluid. For the first time, these physical reductions can be tested. Most importantly, by evidencing a pronounced mismatch in the pressure profile of the shockwave at larger times and high bubble energy, our experimental findings call for an extension of cavitation models.

Overall paper quality: The paper is well written and clear, and I did not spot typos or misprints, which in my reviewer experience is sort of unusual. Figures and related caption are illustrative of the reported work. The relevant technical aspects are well explained and the results appear to be physically sound. Although I do not deem myself a specialist of the field, significant elements of novelty are discussed concerning X-ray holographic data analysis. The proposed technique may potentially allow to obtain significant new data on the cavitation process, at least in the context of single bubble events. The authors claim that their configuration can be (easily?) extended to deal with near wall cavitation, which is of sure interest for many applicative fields, and to address the extreme states of the system related to bubble collapse and plasma dynamics.

Reply: We are pleased that the reviewer finds the MS well written, well explained and clear, and that it may allow for significant new data on the cavitation process. In fact, with the present revision we can already turn this possibility into reality, see also the reply concerning the shortfalls of the Gilmore model above. In order to substantiate our claim regarding the extreme states of the system, we include a graphic on cavitation near a wall in this reply, to convince the reviewer that this is possible, even if we do not want to include it in the MS, since this is too preliminary and will require further beamtime and substantial more theory. Concerning the bubble collapse we can in fact include first results showing bubbles near the collapse (in the revised Supplementary Materials, SM). The experimental challenge here is to achieve a synchronization to probe the collapse ‘point’ itself which we are confident to reach in the beamtime (rescheduled due to the pandemic) based on acoustic stabilization (see Discussion in the revised manuscript and SM).

Fig. for Review 1 ‘Sneak preview image’ image of cavitation events near a wall, showing jetting. In the top row the normal of the interface (glass slide) is along the vertical (slide above of the field-of-view), and jets are hence probed from the side. In the bottom row the interface normal is along the x-ray optical axis, and the jets are projected along the jet axis (center of the bubble). These images are corrected holograms and not yet analyzed by phase retrieval. Preliminary work by J. Rossello et al.

Issues and comments:

- It is well known that single laser-induced cavitation bubbles undergo a characteristic sequence of phases: an initial phase of optical breakdown and plasma formation, a second phase of bubble expansion and a final phase of bubble collapse which, depending on degree of symmetry and bubble energy, may lead to successive rebounds. At collapse, a shock wave is launched in the surrounding liquid environment. This is the shockwave related to cavitation dynamics. A shockwave is also associated with the breakdown phase. From the discussion and from figure 3 and 4.c the authors are seemingly dealing with the breakdown shockwave. The impression is that the text would gain in clarity if an explicit comment is added to explain the issue.

Reply: We have added an explicit comment that we consider the breakdown shockwave.

- Concerning laser induced bubbles, a crucial point is the symmetry achieved in the system. The latter depends on plasma shape and fragmentation, elongation along optical axis and asymmetry with respect to the pump laser focal point. The authors should qualify their data by showing and discussing the plasma images captured from the side-view optical camera.

Reply: We have now added a figure showing an image series recorded with the high-speed camera which demonstrates the high level of shape control achieved. We have added a discussion of the analysis of the optical images in terms of plasma shape.

- The authors introduce an ellipticity factor in the formula used to convert radial phase into radial density profile. It would be nice to correlate this effective parameter, used to prevent the reconstructed density from decreasing below vapor pressure, with the shape of the mature bubble. In fact, any initial lack of sphericity is amplified during the bubble collapse phase making it easy to quantify with the side camera.

Reply: Based on the optical camera images, we can see that most bubbles are quasi-spherical (at the times probed by the high speed camera. However, due to the exposure time of $1\mu\text{s}$ we cannot determine an exact ellipticity value at early bubble times, other than by fixing it from the constraint that the electron density in the bubble center must be close to zero, and in particular cannot be negative. From this criterion we can see that the bubble quickly reaches quasi-spherical shape, i.e. that shape distortions (e.g. in terms of spherical harmonics) are not amplified but instead are damped. Deviations from sphericity at later bubble states (maximum expansion/collapse/rebound) observed with the optical camera can identify but not quantify the asymmetry of the initial breakdown state.

- From the point of view of cavitation, a central point to discuss is the bubble collapse, entailing, if simulations are not mistaken, a strongly focused compression wave and, possibly, transition of the vapor to supercritical conditions. In principle, as claimed by the authors, the proposed set-up should enable to follow this phase. Is there a reason why the authors do not show data for the collapse? Did they encounter substantial difficulties in dealing with that? In the present referee view point, adding even partial data on this aspect would crucially demonstrate the strength of the approach and substantially enhance the value of the paper.

Reply: The main challenge in capturing the bubble collapse is to achieve the temporal overlap, since the jitter of the collapsing bubble is too large. In other words, by synchronisation of the IR and X-ray laser alone, we can capture the extreme point of collapse only with extreme luck (one lucky shot per ~ 2 hours). Therefore, the cavitation bubble has to be acoustically trapped

and this ultrasound field has to be synchronized in addition. We have already worked out a scheme for this task and had been granted MID/XFEL beamtime to study the acoustically stabilized cavitation bubble collapse, but this beamtime was cancelled due to the pandemic. We have, however, now included our first images of near-collapse recordings shortly before and after the collapse (rebound) in the SM.

- Do the authors believe their technique could be extended to axisymmetric bubbles? Ideally, the bubbles are more likely to be axisymmetric along the optical axis of the pump laser beam rather than spherical. I imagine that, by taking holograms from the side, more or less the same analysis technique should work except that a cylindrical rather than spherical Abel transform would be needed to reconstruct the bubble interior. Unless I am wrong, the authors should explicitly comment on this important issue (see also the last item of the present list).

Reply: Axisymmetric bubbles, recorded with the long axis along the projection are already included when taking the asphericity/ellipticity parameter into account. We have not performed recordings yet on axisymmetric bubbles with the laser optical axis perpendicular to the X-ray optical axis which is an excellent idea. This case is fully contained in the holography analysis workflow, after adapting the integration and propagators or using the 2d AP phase retrieval approach. On the reconstructed phase projections, the cylindrical inverse Abel transform would have to be applied for each vertical line of the image. This is now explained.

- The study of extreme states associated with optical breakdown, plasma formation, bubble collapse/rebound and shock wave dynamics are presented as perspective applications, as suggested by the sentence *An increase in the frame rate of detectors used for ultra-fast holographic imaging to kHz would enable . . .* In order to resolve the implied fast dynamics, an extremely fast sampling rate is mandatory. The author should illustrate more in depth the available detector technology and related limitations and discuss how far is the state of the art from achieving the required resolution.

Reply: There are indeed very interesting detector developments leading to MHz frame rates also for small pixel detectors (see ref. [26]), but the time scales here can only be covered with pump-probe techniques so that an ample discussion on detector technology would be misleading.

- The Fresnel number of the optical system is given as $F = 7.6 \cdot 10^{-4}$ which, according to classical definitions, should correspond to the far-field, Fraunhofer regime. It is then not clear why is the approach referred to as near-field holographic imaging. Am I missing something?

Reply: The approach is denoted as near-field since the image is formed by superposition of scattered and (divergent) primary beam. While F defined with respect to a single pixel is already well below one, the Fresnel number defined with respect to object and beam size is significantly larger. Importantly, the Fresnel propagator and not the Fourier operator is required to compute the signal.

- The proposed analysis is deeply rooted on the assumption of spherical symmetry. On the contrary near wall cavitation bubble dynamics is never spherically symmetric, implying the formation of high speed jets toward or away from the boundary, for solid walls and free surfaces, respectively. In order to avoid conjectural sentences and for the interest the issue may have, can the authors explain how they plan to extend their approach to such much more compli-

cated geometrical conditions?

Reply: In the future, we plan recordings with several X-ray beams illuminating the sample at an angle. Beam splitters based on X-ray waveguide optics have already been tested by our group and may provide one option of implementation, but we may also use grating based beam splitters and two sets of CRLs. We could then use the information of stereo or multiple views (waveguide beamlets), in combination with a reasonable modelling of the bubble shape, e.g. in terms of spherical harmonics expansion. For the immediate future (next beamtime) we will simply shoot the IR laser at 90° to the x-ray optical axis so that we probe perpendicular to the axis of symmetry, as suggested by the reviewer above.

Recommendation: In conclusion, I read the paper with sincere interest and I understood the reported novelty as being related more to the field of X-ray holography rather than to bubble dynamics and cavitation. I believe that the authors should make an additional step toward actually demonstrating the potential of their new holographic technique for the specific field of single cavitation bubble dynamics by addressing the comments reported in the list above in view of the eventual publication of their paper in Nature Communications.

Reply: We thank the reviewer for his/her helpful suggestions and are confident that he/she considers the substantial extensions performed in this revision as the expected 'additional step', in view of a full recommendation for publication. This concerns in particular the computational fluid dynamics modelling and the corresponding comparison of the pressure profiles. As a quantitative result on an observable which was so far experimentally inaccessible, the spatial pressure distribution provides substantial novelty concerning the understanding of cavitation!

Reviewer: 2

Reviewer #2 (Remarks to the Author):

The authors present a time-resolved imaging study of laser-induced cavitation bubbles in water at the micrometer and nanosecond time scales. The setup is very elaborate, the data analysis is detailed and well explained. However, I would recommend to reject the publication of this manuscript in Nature Communications due to insufficient justification of general interest. The authors suggest that their paper presents a new imaging method for shock waves with significant advantage over previous visualization attempts in spatial and temporal resolutions(1), and that their method delivers new insights into the evolution of electron density in shock fronts, which can be used to improve and calibrate existing theoretical models (2). In my opinion, at least one point (1) or (2) have to be fulfilled for publication in Nat. Comm, but the authors don't present enough evidence to support this impression.

Concerning point 1:

1a. Imaging of shock fronts with XFEL using holography has been successfully demonstrated in diamond and other materials in Schropp, Andreas, et al. "Imaging shock waves in diamond with both high temporal and spatial resolution at an XFEL." Scientific reports 5 (2015): 11089. This study is not, but should be cited in this manuscript. The imaging setup, and the temporal and spatial resolutions were very similar to the ones presented in the current study. Some variations in image analysis is the most prominent difference at the first glance.

Reply: We thank the reviewer for pointing to the reference on x-ray imaging of laser induced shock waves in diamonds. The paper is now included and indeed gives a good starting point for understanding the novelty in our setup. Notably, the image quality and flawed empty beam division previously impeded quantitative analysis of the phase/electron density, which we have now achieved. Furthermore, the dynamics imaged in the present work contains phase transitions and phase boundaries in liquid as well as a nearly spherical geometry of the shockfront, while the cited work is concerned with single phase and plane shocks. We consider this as a quite significant advance.

1b. Also, it is true that the spatial resolution of near field X-ray holography is not limited by the detector pixel size in contrast to X-ray microscopy. However, it is not an accident that both methods achieve similar spatial resolutions. In most imaging experiments the resolution limit is determined by the brightness of the image features which carry the high spatial resolution information. This factor is mostly determined by the FEL pulse energy, which is not really scalable (in contrast to the pixel angular size). This is why it will be hard to image sub-micrometer bubbles with tens of nanometers resolution using the presented method. As such, both methods face similar limitations and near-field X-ray holography does not present a significant scalable improvement.

Reply: We are pleased, that the reviewer acknowledges that the spatial resolution of near field X-ray holography is not limited by the detector pixel size. We must politely refute the statement, that the resolution of X-ray holography and X-ray microscopy using a high resolution scintillator microscope as in ref. [26] exhibits the same resolution. This is not the case here, nor in general, as we have shown in numerous work with synchrotron radiation, see for example Bartels et al. Phys. Rev. Lett 2015. <https://doi.org/10.1103/PhysRevLett.114.048103>. There is no reason why scalability should not hold for FEL single pulses. The total fluence of a single pulse is larger than that used in our synchrotron exposures. Further information on X-ray holography resolution and scaling thereof is also found in:

Salditt T., Töpperwien M. (2020) Holographic Imaging and Tomography of Biological Cells and Tissues. In: Salditt T., Egner A., Luke D. (eds) Nanoscale Photonic Imaging. Topics in Applied Physics, vol 134. Springer, Cham. https://doi.org/10.1007/978-3-030-34413-9_13

Salditt T., Robisch AL. (2020) Coherent X-ray Imaging. In: Salditt T., Egner A., Luke D. (eds) Nanoscale Photonic Imaging. Topics in Applied Physics, vol 134. Springer, Cham. https://doi.org/10.1007/978-3-030-34413-9_2

1c. Even a greater problem of the manuscript is that it is not well explained why the presented study can deliver results which differ greatly from high-resolution and high speed optical imaging. Especially for medium-sized bubbles in early stages of the expansion multi-scattering effects seem not to be very pronounced. The experimental work cited in the publication is mostly over 20 years old.

Reply: The fact that most studies are decades old are also indication that the optical imaging has already been driven to the limits, and not much advance could be achieved in the optical domain in the meantime. Here, in particular the work by Vogel et al still sets the standards.

For example, newer studies such as Charee, Wisan, and Viboon Tangwarodomnukun. "Dynamic features of bubble induced by a nanosecond pulse laser in still and flowing water." Optics & Laser Technology 100 (2018): 230-243. nicely visualize shock formation at the

boundary of bubbles. This study, which relies on a much cheaper direct imaging method, exhibit a similar clarity of detail without the artifacts of inline X-ray holography such as ringing and the need of reconstruction algorithms.

Reply: We are sorry, but we could not find any shock wave imaging in the indicated paper, not to speak of a sequence with high temporal or spatial resolution. The paper is concerned with laser ablation, i.e. production of solid debris (e.g. nanoparticles) from solid targets under water and/or cutting/shaping of a workpiece by laser pulses. The study investigates the dynamics of bubbles occurring in the ablation process under high repetition rates of the laser pulses, which is quite different from our setup (individual single bubbles in the bulk liquid). The imaging temporal resolution amounts to $10\mu\text{s}$, which is more than 3 orders of magnitude less than in our experiment. The scale bar, which is tiny in the optical micrograph corresponds to $100\mu\text{m}$, and the pixel size can be calculated to be $\sim 3\mu\text{m}$, not to speak of the resolution. The exposure time is not mentioned, but the blurring of bubble walls suggests that it was about several μs as well. Shock waves near the expanding or collapsing bubble cannot be imaged with this timing. We have now scanned again the recent literature on laser induced cavitation and shock wave measurements. We did not find any recent addition or update to the older results with respect to our setup (most publications are dealing with laser ablation). However, we updated our reference list and included now several papers on this topic, in particular the review by Lauterborn and Vogel (2013) that still gives a good overview on the state-of-the-art. Latest reports on Streak imaging experiments achieve very high temporal resolution but do not allow quantitative analysis of density or pressure. See refs [11-13].

Concerning point (2): In the introduction, the authors claim that high resolution X-ray imaging will help to understand and better simulate the formation of bubbles and electron density of shock fronts. However, in the discussion section there is no mentioning if the results are a surprise to or align well with theoretical predictions. Overall, it seems that the process the authors observe is very well understood. If this is not the case, the authors have to stress this point more vigorously.

Reply: The comment of the reviewer has prompted us to carry out a substantial further analysis of the data, and to exploit the unique advantage of our method providing an independent measurement of the pressure distribution (and thus the shock pressure) via the electron density (i.e. liquid density) around the expanding bubble. The method does not rely on the shock front speed nor on hydrophone data far from the expanding bubble, and hence gives insight into cavitation phenomena provided by no other technique before.

In particular we now show that the standard cavitation model according to Gilmore (quasi-analytical model of spherical cavitation bubble) fails to describe the pressure profile at high bubble energies. For the first time, pressure in the shockwave is not only determined from the shock wave velocity based on assumptions and literature values for the equation-of-state, but can independently be measured. The discrepancy between data and model now motivates a major reinvestigation of the basic theoretic assumptions involved in the model: the Kirkwood-Bethe hypothesis, and an assumption on the laser seeding process. We have included a new figure on these new results, as have completely rewritten the discussion. All required details on the computational fluid dynamic models are given in the main text, or the SM, which also has been substantially extended.

Importantly, the results now presented meet the criteria (2) laid out by the reviewer for a positive recommendation. We are confident that publication will be endorsed.

Finally, we want to thank both reviewers who have helped us to substantially improve the manuscript. In fact, we enjoyed investing several weeks in this revision, forcing us to push the analysis and exploitation of the data directed at a more quantitative understanding of cavitation dynamics.

Reviewers' Comments:

Reviewer #1:

Remarks to the Author:

I have read the revised version of the manuscript and found that the new version has been considerably enriched in terms of physics of cavitation. In particular, the comparison with numerical simulations using the more or less standard Gilmore model is interesting, showing that certain quantities are well captured by the current models and others are not (the pressure profile at high bubble energy in particular). Many more details and interesting data are provided in the supplementary materials which complement the main findings discussed in the main text.

I appreciated the effort put by the authors in providing detailed answers to all the issues and questions I was raising in my original report. I am grateful for confidentially sharing in the rebuttal letter their preliminary results concerning bubble dynamics and jetting close to a solid wall. These data convincingly showed that their claims about the potential extension to non spherical conditions are actually realizable. Also interesting are the (partial) results on the bubble collapse phase illustrated in the Supplementary Materials.

In conclusion, the proposed approach has the great potential of measuring the spatial density distribution inside the bubble and the shock wave region, which is a feature not achieved by other, more traditional, experimental approaches. This can be considered a potentially important advancement with respect to the current experimental state of the art. In fact, as the authors comment, the current approaches are sort of frozen by more than a decade suggesting that they have already been fully exploited and that novel proposals are needed.

Though I am still convinced that the authors did not fully extract all the physical information from their data, I believe the approach they propose is new and able to provide new insight in the field of cavitation. The latter, though probably being a quite focused research field, is deeply connected to many general open problems in physics and technology.

In conclusion, on my part, I am convinced that, in the present revised form, the paper fully merits to be published in Nature Communications.

Carlo Casciola

Reviewer #2:

Remarks to the Author:

First, I will start with some positive remarks. The authors have improved the manuscript and were responsive to most comments. Two main points now can be crystallized from the manuscript. First, the authors claim that one of the main advantages of their method is that they can image the density distribution within the shock front with high spatial resolution, which is difficult with optical and other methods. Second, after a second look the authors claim to see effects which cannot be explained with existing theory.

However, in my view, the paper is still filled with inaccurate and misleading statements which

unfortunately do not support the case for publication. For example, the paper repeatedly claims that they have achieved unprecedented spatial and temporal resolution. However, there is no clear discussion about the actually achieved spatial resolution which is usually worse than the nominal resolution. When comparing the optical images from other publications (see the foils 1-3 attached to the email) and the X-ray images the difference does not seem dramatic. Besides most optical images show less ringing effects and do not require pre-assumptions for reconstruction. Specifically the critical region near the shock front in X-ray holograms could suffer from ringing effects and other artefacts brought by coherent imaging which you can see in the reconstruction image in Fig 2, d.

The other advantage of the optical measurements are truly stroboscopic as they can follow the evolution of a single bubble. In the current publication the bubble is being destroyed during every imaging event and the dynamics have to be indirectly reconstructed from many different bubbles.

Another claim is that the method allows for: "... extracting the 3d-pressure distribution of individual shockwaves in space and time...". However, this is only true for full spherical symmetry which is clearly not given for higher excitation laser pulse energies as admitted by the authors and shown in foil 4. That is problematic when the authors claim that they find an interesting discrepancy between the theory using spherical symmetry and the experiment. The authors repeat that "... However, realistic shape distortions could not explain the inversion of the pressure slope between simulation and data...", but I don't find a convincing argument why that is the case. As can be seen in foil 4, the shape of the bubble and the shockfront can greatly deviate even from an ellipse. Also for some reason the time delays for the experiment and simulation are quite different $t = 2 \text{ ns}$, 5 ns and 15 ns , and $t = 1.4 \text{ ns}$, 6.5 ns and 13.3 ns . Besides, the differences in pressure and thus density are significant over 10-20 micrometers and I am a bit surprised if these features would not appear in the optical images recorded with modern methods such as demonstrated in D.Veysset et al., PRL E 97 (foil 1-2).

In summary, that present work may be describing interesting insights but the paper still needs significant additions to clearly demonstrate this.

NCOMMS-20-28821-T /Revision 2:

Response to Reviews & Summary of Changes

Reviewer 1

Reviewer #1 (Remarks to the Author):

I have read the revised version of the manuscript and found that the new version has been considerably enriched in terms of physics of cavitation. ...I appreciated the effort put by the authors in providing detailed answers to all the issues and questions I was raising in my original report. ... In conclusion, the proposed approach has the great potential of measuring the spatial density distribution inside the bubble and the shock wave region, which is a feature not achieved by other, more traditional, experimental approaches..... In fact, as the authors comment, the current approaches are sort of frozen by more than a decade suggesting that they have already been fully exploited and that novel proposals are needed The latter, though probably being a quite focused research field, is deeply connected to many general open problems in physics and technology. ...In the present revised form, the paper fully merits to be published in Nature Communications. Carlo Casciola

Reply: We thank Dr. Casciola, who is so highly reputed in fluid dynamics, for his careful review and positive assessment.

Reviewer: 2

The authors have improved the manuscript and were responsive to most comments. Two main points now can be crystallized from the manuscript. First, the authors claim that one of the main advantages of their method is that they can image the density distribution within the shock front with high spatial resolution, which is difficult with optical and other methods. Second, after a second look the authors claim to see effects which cannot be explained with existing theory.

Reply: We thank the reviewer for acknowledging the improvement and for prompting us not to content ourselves with demonstrating a new imaging modality, but to really exploit the full quantitative capability of our method.

However, in my view, the paper is still filled with inaccurate and misleading statements which unfortunately do not support the case for publication. For example, the paper repeatedly claims that they have achieved unprecedented spatial and temporal resolution. However, there is no clear discussion about the actually achieved spatial resolution which is usually worse than the nominal resolution. When comparing the optical images from other publications (see the foils 1-3 attached to the email) and the X-ray images the difference does not seem dramatic.

Reply: We do not claim our resolution to be a ‘dramatic’ improvement with respect to optical resolution, in general (i.e. $\lambda/2$). But it is important to note, that optical imaging of cavitation does not achieve the $\lambda/2$, due to the restrictions in numerical aperture (cuvette size, window materials etc.). Best effort optical imaging of cavitation bubbles (in bulk water cavitation) yields a resolution which- to best of our knowledge - is lower by a factor 8. We have now performed a full quantification of our resolution, including the important step of phase retrieval, and have added a corresponding section with an additional compound figure in the SM. Accordingly, the resolution is currently below 500nm, and hence – as the reviewer correctly assumed - does not

yet reach the effective pixel size and nominal focal spot size (around 100nm). As we now explain in detail, this can be attributed to shortcomings in longitudinal coherence of the SASE pulse at current machine operation in combination with the dispersive optical focusing scheme (CRL). Importantly, we now also describe how this can be circumvented by either seeded SASE radiation or the use of non-dispersive focusing optics, which is routinely achieved for holography with synchrotron radiation with resolutions better than 30nm (e.g. Bartels et.al. PRL 2015, ref. 3 of SM). The present resolution of 0.5 μ m is, however, fully sufficient for the conclusions drawn, in particular since fluid dynamics and surface tension smoothen the shape of the cavitation bubble. More importantly, the resolution can be easily scaled up in the next beamtime, without the limits which optical imaging is facing (NA- and wavelength-associated), as we now explain in the revised SM. This is especially important for future experiments on the bubble collapse, where cuvettes with dimensions larger than 1cm are needed for unconstrained bubble expansion, limiting the NA of optical imaging.

Importantly, with the present experiment we are already about a factor of 4-8 better than optical imaging of cavitation bubbles, see the direct comparison in the Figure below, including the optical work referred to by the reviewer.

[Redacted]

Figure 1. Comparison optical / X-ray. Reproduction of optical micrographs mentioned by the reviewer: (a) Lauterborn and Vogel (reviewer attachment 2), (b) Veysset et al., attachment 1), (c) Kim et al (attachment 3), compared to (d,e) present work (upper left: hologram, lower right: reconstruction). Panels (a-d) are shown to scale, and also represent comparable time delays. Scale bars (a) 100 μ m, (b,c) 50 μ m, (d,e) 25 μ m. The resolution is quantified in the revised SM.

We have now precisely detailed and discussed the resolution. For single pulse imaging, the acclaimed work of Vagovic et al. Optica 2019 (ref. 30), showing exploding glass with an effective pixel size of 3.2 μ m can in fact mark the current benchmark of fullfield XFEL imaging. Pulse-limited holographic full-field X-ray imaging with sub-micron resolution can hence be considered as a novelty. Further, the resolution of the present approach could easily be scaled up by a factor of 10 by changing from a dispersive to a non-dispersive x-ray optics (KB-mirrors, waveguides), as we now discuss in the supplemental material.

Besides most optical images show less ringing effects and do not require pre-assumptions for reconstruction. Specifically the critical region near the shock front in X-ray holograms could suffer from ringing effects and other artefacts brought by coherent imaging which you can see in the reconstruction image in Fig 2, d.

Reply: The coherent ringing in the hologram represents the signal, not an artefact. Phase retrieval inverts the information in these fringes and yields the projected phase. For insufficient phase retrieval, some residual fringes can be observed, but the RFP algorithm which we used for the analysis (in contrast to the conventional AP which we use only for illustrative purposes) does not suffer from strong residual artefacts. This is now better explained.

The other advantage of the optical measurements are truly stroboscopic as they can follow the evolution of a single bubble. In the current publication the bubble is being destroyed during every imaging event and the dynamics have to be indirectly reconstructed from many different bubbles.

Reply: Observing the full dynamic cycle of a single cavitation bubble is of course interesting and helpful. In fact, we did image the whole bubble cycle of each individual cavitation event with an optical high-speed camera, in parallel to the X-ray measurement. The present experiment, however, did aim at imaging the first 100ns of the cavitation bubble dynamics after dielectric breakdown. For these early states, many optical high-speed measurements also rely on a pump-probe scheme with newly seeded bubbles for each shot (e.g. Vogel et al. J. Acoust. Soc. Am. 1996, Toker et al. Chem. Phys. Lett. 2009, Potemkin et al. Laser Phy. Lett. 2014 or Veysset et al. Scientific Reports 2016). Multiframe high-speed imaging of single cavitation bubbles with optical contrast has been demonstrated e.g. in Tagawa et al. J. Fluid. Mech. 2016, Veysset et al. Phys. Rev. E 2018 and Kim et al. Science Advances 2020 but can only inform on bubble and shock wave radius and shape, not on the density profile. All papers are cited and discussed in the manuscript.

Another claim is that the method allows for: "... extracting the 3d-pressure distribution of individual shockwaves in space and time...". However, this is only true for full spherical symmetry which is clearly not given for higher excitation laser pulse energies as admitted by the authors and shown in foil 4. That is problematic when the authors claim that they find an interesting discrepancy between the theory using spherical symmetry and the experiment. The authors repeat that "... However, realistic shape distortions could not explain the inversion of the pressure slope between simulation and data...", but I don't find a convincing argument why that is the case.

Reply: We agree with the reviewer that this needed more explanation. We have added a full section in the SM showing by analytical and numerical simulations that realistic shape distortions only change the scaling of the extracted profile, not its form. Note as well that the shape deviations are to some extent also controlled by the orthogonal camera, see the new section on vetoing out multiple plasmas, where this is now discussed. Most importantly, moderate shape distortions scale the profile, but do not change its form, as we now show by simulations in a new section of the supplemental material. We have also included a discussion of linear stability analysis (added references in the SM), which show that shape deformations are damped for expanding bubbles at our experimental parameters.

As can be seen in foil 4, the shape of the bubble and the shockfront can greatly deviate even from an ellipse.

Reply: At the regime of sub-threshold laser intensity used in this work, the strongly elongated plasma shapes, which the reviewer has in mind from the work of Vogel and Lauterborn J. Acoust. Soc. Am. 1996 are very unlikely (in contrast to seeding with higher pulse energy)

Also for some reason the time delays for the experiment and simulation are quite different $t = 2$ ns, 5 ns and 15 ns, and $t = 1.4$ ns, 6.5 ns and 13.3 ns.

Reply: The simulation has four effective parameters for the initial conditions: Pulse duration, seeding time, bubble energy (from laser pulse energy) and plasma core radius at time zero. These conditions shift the effective initial time point. Comparing to the experiment, we therefore matched the propagation of the shock wave and the bubble wall, not the nominal time which depends on the parameters and the exact shape of the laser pulse.

Besides, the differences in pressure and thus density are significant over 10-20 micrometers and I am a bit surprised if these features would not appear in the optical images recorded with modern methods such as demonstrated in D.Veysset et al., PRL E 97 (foil 1-2).

Reply: This is precisely since the density cannot be properly imaged. The optical contrast is dominated by the white plasma (light emission) and black bubble (light reflection/refraction). The increased density of the shock wave does not come out properly, not even in qualitatively, let alone quantitatively. Only the gradient of density at the outer shockwave radius results in some contrast, which allows one to determine the shock wave radius, but not its density.

In summary, that present work may be describing interesting insights but the paper still needs significant additions to clearly demonstrate this.

Reply: We have now provided the requested additions, and thank the reviewer for prompting us in particular to expand on the important issue of the projection of deformed bubbles. We also are confident that our added explanations now have removed any unclarities and misconceptions on the unique nature of the contrast provided here and the resolution advantage.

Reviewers' Comments:

Reviewer #2:

Remarks to the Author:

All my comments were addressed by the authors, I think that the authors clearly communicate the advantage and the potential of their method. Overall I recommend this manuscript for publication in Nat. Comm. with a minor revision.

There is still the issue with the spherical assumption which is still not resolved in my eyes. The authors present their answer in the last figure of the supplements subsection, which assumes a core-shell model with two constant densities in the shell and the core. It is obvious that in such a simplified case the 2D projection is similar for different shapes, that was not the point. The issue here is that there is no clear reason why the shock wave density distribution along the laser direction should be homogeneous as suggested in S6. In fact optical data suggests that this is not the case as shown in Figure 1a (answer to the referees). The shock front from the bubble cone tip is further advanced than the cone bottom. If the shock fronts move at different velocities and have different density distributions, their relative density differences would simply sum up in the 2D projection and could show up as deviation from a spherical assumption. The slow optical camera cannot exclude such interesting early stages in my opinion.

I think that the symmetry issue can be resolved if the authors just mention that inhomogeneities along the laser direction cannot be excluded and may play a significant role (or explain why they can be excluded if I misunderstood something). In fact, this might be even a motivation for an outlook study where the optical laser is perpendicular to the XFEL. Temporal overlap issues will make the experiment more difficult, but maybe it is worth it and will emphasize the capabilities of this new approach. I also recommend to remove figure S6, I think the figure is not helpful for understanding the approach and rooted in a misunderstanding in our discussion.

NCOMMS-20-28821-T /Revision 3 (Minor):

Response to Reviews & Changes Made

REVIEWERS' COMMENTS reviewer #2 (Remarks to the Author):

All my comments were addressed by the authors, I think that the authors clearly communicate the advantage and the potential of their method. Overall I recommend this manuscript for publication in Nat. Comm. with a minor revision.

Reply: We thank the reviewer for the help and advice provided, in particular in improving a clear communication of the method's advantages and potential, and are very grateful for his/her recommendation. It is a pleasure to provide the final minor revision.

There is still the issue with the spherical assumption which is still not resolved in my eyes. The authors present their answer in the last figure of the supplements subsection, which assumes a core-shell model with two constant densities in the shell and the core. It is obvious that in such a simplified case the 2D projection is similar for different shapes, that was not the point. The issue here is that there is no clear reason why the shock wave density distribution along the laser direction should be homogeneous as suggested in S6. In fact optical data suggests that this is not the case as shown in Figure 1a (answer to the referees).

The shock front from the bubble cone tip is further advanced than the cone bottom. If the shock fronts move at different velocities and have different density distributions, their relative density differences would simply sum up in the 2D projection and could show up as deviation from a spherical assumption. The slow optical camera cannot exclude such interesting early stages in my opinion. I think that the symmetry issue can be resolved if the authors just mention that inhomogeneities along the laser direction cannot be excluded and may play a significant role (or explain why they can be excluded if I misunderstood something). In fact, this might be even a motivation for an outlook study where the optical laser is perpendicular to the XFEL. Temporal overlap issues will make the experiment more difficult, but maybe it is worth it and will emphasize the capabilities of this new approach. I also recommend to remove figure S6, I think the figure is not helpful for understanding the approach and rooted in a misunderstanding in our discussion.

Reply: We agree that the shock wave density must not be homogeneous in the laser direction. Indeed, Figure S6 covers a different and arguably simpler scenario where the profile function of the shockwave is homogenous but then imposed on a distorted shape. The robustness of the projection analysis and reconstruction then applies also to profiles with non-constant shape along the interface normal. However, inhomogeneous densities of the shockwave when moving parallel to the surface of the bubble are indeed more problematic. We now mention this very clearly in the main text and in the presentation of S6, thus following the reviewer's suggestion to resolve this issue. We now also state explicitly that the slow optical camera cannot exclude such interesting effects in the early stages. We did decide to keep Figure S6, however, since it informs on a case which is also important and insightful. Misunderstandings are now avoided by the rephrased presentation. And, finally, as the reviewer points out, this provides a perfect 'motivation for an outlook study where the optical laser is perpendicular to the XFEL' – as we now propose in the outlook.